# PROT2RNA: A DIFFUSION LANGUAGE MODEL FOR PROTEIN-CONDITIONED MRNA CODING SEQUENCE GENERATION

## ABSTRACT

The redundancy of the genetic code, where multiple codons encode the same amino acid, creates a vast design space for messenger RNA (mRNA) sequences. Synonymous codon choices significantly affect mRNA stability, structure, translation efficiency, and immunogenicity, all critical for mRNA therapeutics and synthetic biology. We present Prot2RNA, a diffusion language model that generates mRNA coding sequences conditioned on a target protein. Prot2RNA uses a two-stage training approach: the model is pretrained using masked diffusion modeling over separate sets of human protein and mRNA coding sequences, learning representations for both biological modalities in a shared space. Subsequently, the model is finetuned to generate codon sequences using target protein sequences as prompts. Prot2RNA was trained on human data and evaluated on a held-out set of highly expressed mRNA transcripts that are sequentially different from the training set. The results demonstrate that our diffusion-based codon optimization model outperforms existing methods in codon-level accuracy, alignment with biologically meaningful properties, and its ability to generate sequence profiles that closely mirror codon usage patterns in highly expressed wild-type human mRNAs. Unlike other deep learning models that primarily learn codon usage frequency, Prot2RNA implicitly learns biologically relevant codon preferences, providing a strong foundation for protein-aware mRNA design.

## 1 INTRODUCTION

Messenger RNA (mRNA) has rapidly emerged as a powerful modality for gene therapy (Chehelgerdi & Chehelgerdi, 2023), antibody and protein replacement (Deal et al., 2023; Vavilis et al., 2023), personalised cancer vaccines (Tadic & Martínez, 2024), and mRNA vaccines (Hussain et al., 2022). By delivering a nucleotide blueprint rather than a protein payload, mRNA enables both rapid manufacturing and *in situ* protein expression (Leong et al., 2025; Yingying Shi & You, 2024). However, mRNA therapeutics still face important limitations, particularly suboptimal expression of the encoded payload. Addressing this challenge is essential to ensure success in preclinical and clinical applications (Sahin et al., 2014; Jackson et al., 2020; Crommelin et al., 2021).

The efficiency of mRNA translation depends heavily on the design of its coding sequence (CDS), where the choice of synonymous codons, referring to different codons that encode the same amino acid, can significantly impact protein output. Due to redundancy of the genetic code, where 61 codons encode just 20 amino acids, a single protein can be encoded by an enormous number of alternative mRNA sequences that differ in codon composition but produce the same amino acid chain. For example, the $1,273$-amino-acid SARS-CoV-2 spike protein can be encoded in more than $10^{632}$ different synonymous ways (Kim et al., 2024). However, only a small fraction of these sequences lead to high expression in human cells, a stable secondary structure, and favorable translational kinetics (Walsh et al., 2020; Moss et al., 2022; Alevras et al., 2023). Identifying such high-performing sequences remains a major computational challenge.

Numerous strategies have been developed to optimize codon composition within the CDS, aiming to improve mRNA translation efficiency and stability. Traditional approaches prioritize replacing rare codons with those more frequently used in the host organism, using metrics like the Codon Adapta-

tion Index (CAI) to correlate codon usage with protein expression levels (Sharp & Li, 1987). Other methods incorporate stability-focused metrics, such as the Codon Stabilization Coefficient (CSC) (Presnyak et al., 2015a), or structural considerations, such as pseudo-MFE models that reduce overly stable regions within the coding sequence (Gaspar et al., 2013). These strategies reflect the multifactorial nature of codon optimization, but most rely on isolated metrics or simplified assumptions, which may fail to capture the complex dependencies that shape protein output. To address these limitations, recent work has turned to deep learning models that learn codon-expression relationships directly from data.

Although deep learning has enabled more expressive models for codon optimization, current approaches still face key limitations. Many are trained on cross-species training corpora, which may dilute human-specific codon usage patterns critical for therapeutic design. Others optimize predefined objectives such as CAI or folding energy, which do not consistently align with high expression in human cells (Vogel et al., 2010). Additionally, evaluation is often performed on randomly heldout sequences that are similar to the training data, raising concerns about their performance on novel protein sequences.

In response to these limitations, we introduce Prot2RNA, a diffusion language model (DLM) for codon optimization, trained exclusively on human coding sequences. Our goal is to learn expression-relevant codon preferences directly from human data, without relying on handcrafted metrics or signals from other species. To assess generalization, we construct a rigorous benchmark of highly expressed human mRNA coding sequences that are sequentially dissimilar from the training set. Our model generates synonymous sequences with codon usage patterns closely matching those of highly expressed human transcripts, achieving strong codon-level accuracy and biologically plausible generalization to novel proteins.

The main contributions of the paper are as follows:

- Collected and preprocessed human protein-mRNA pairs dataset for codon optimization model training and benchmarking;[1]
- A set of publicly available DLMs for codon optimization, pretrained and finetuned on protein and mRNA coding sequences;[1]
- Extended numerical experiments showing the superior performance of DLMs in generating coding sequences that preserve fine-grained positional codon trends found in highly expressed natural transcripts.

## 2 RELATED WORK

**Language Models for Biological Sequences** Transformer-based language models have become the cornerstone of biological sequence modeling. Models such as ESM1b (Rives et al., 2021) and ESM-2 (Lin et al., 2023) use masked language modeling to capture evolutionary and structural patterns, while generative approaches like ProtGPT2 (Ferruz et al., 2022) support *de novo* protein design. More recent efforts, including DPLM-2 (Wang et al., 2025) and ESM3 (Hayes et al., 2025), integrate both sequence and structure information to improve performance in various tasks.

In parallel, using similar training objectives, RNA language models have been developed primarily for noncoding RNAs. Examples include RiNALMo (Penić et al., 2025), RNA-FM (Chen et al., 2022), and RNA-MSM (Zhang et al., 2023b), which learn representations of RNA structure and function. Recent generative models such as GenerRNA (Zhao et al., 2024), GARNET (Shulgina et al., 2024), and RfamGen (Sumi et al., 2024) focus on the design of structured RNAs, guided by secondary structure or family-level constraints. While not directly applicable to coding RNA, these approaches highlight strategies that may benefit future models for thermodynamically stable mRNA design.

**mRNA Predictive Models** Several recent models have shifted attention toward coding RNAs, capturing determinants of mRNA expression, stability, and translation efficiency. Typically using encoder-only architectures pretrained on millions of coding sequences, examples include Sanofi's

---

[1]Currently available at `https://figshare.com/s/414056ccf253acb31a4a`; will be made public upon acceptance.

CodonBERT (Li et al., 2024), UTR-LM (Chu et al., 2024), and mRNA-LM (Li et al., 2025). While these models reveal important insights into mRNA function, they mainly score or rank existing sequences rather than generate optimized ones, leaving sequence generation an underexplored challenge.

**Models for Codon Optimization**   In contrast, generative models for codon optimization aim to synthesize new coding sequences that maximize expression efficiency. These models vary substantially in architecture, supervision, and underlying biological assumptions. CodonBERT (Ren et al., 2024) adopts a simple cross-attention mechanism to model alignment between amino acids and codons. In contrast, CodonTransformer (Fallahpour et al., 2025) jointly encodes protein-codon pairs using a custom STREAM tokenization scheme and BigBird encoder, using the classical masked language modeling objective. Trias (Faizi et al., 2025) introduces a vanilla generative encoder-decoder architecture using species-specific tokens, enabling protein-to-CDS generation across species. These models learn codon preferences implicitly from coding sequences but differ in how they represent the protein-RNA relationship. GEMORNA (Zhang et al., 2024) goes further and generates both CDS and UTRs using modular Transformer models, however, the model is not publicly available. Beyond neural models, LinearDesign (Zhang et al., 2023a) offers an alternative by using lattice parsing and a finite-state automaton to optimize both codon usage and RNA stability jointly. Together, these models illustrate a growing interest in data-driven mRNA design, with varied approaches targeting organism-specific tuning, sequence interpretability, and translational effectiveness.

**Diffusion and Iterative Refinement for Discrete Sequences**   A growing body of work extends diffusion and iterative refinement methods to discrete token spaces. Early non-autoregressive refinement approaches such as Mask-Predict (Ghazvininejad et al., 2019), SUNDAE (Savinov et al., 2022), ARDM (Hoogeboom et al., 2022), and MaskGIT (Chang et al., 2022) demonstrated the effectiveness of parallel multi-step denoising for tokens. Subsequent discrete diffusion models—including D3PM (Austin et al., 2021), Multinomial Diffusion (Hoogeboom et al., 2021), LLaDA (Nie et al., 2025), and MDLM (Sahoo et al., 2024), formalize corruption–denoising processes for symbolic sequences. While these approaches provide the algorithmic foundations for discrete multi-step generation, none address protein-conditioned mRNA generation.

# 3 METHODS

## 3.1 PROT2RNA LANGUAGE MODEL

Prot2RNA is a diffusion language model designed for protein-conditioned generation of mRNA coding sequences. It follows the discrete masked-denoising formulation of LLaDA (Nie et al., 2025), but adapts it to a multimodal biological setting where each amino acid aligns with a codon triplet. In this framework, generation proceeds as an iterative denoising process, the model progressively denoises a masked RNA sequence conditioned on a visible protein sequence, updating all masked tokens in parallel rather than decoding left-to-right. This parallel diffusion strategy avoids the arbitrary causal order imposed by autoregressive (AR) models and better captures the bidirectional dependencies and global compositional constraints (e.g., GC balance, codon periodicity) that underlie biological sequences.

Training proceeds in two stages: i) masked diffusion pretraining on individual protein or mRNA coding sequences teaches the model general sequence statistics in a shared token space (Figure 1A), and ii) instruction-style finetuning on protein-mRNA pairs, keeping the protein tokens unmasked while iteratively denoising the corresponding RNA segment (Figure 1B). This setup aligns modalities at the residue-codon level and enables protein-conditioned generation within a single unified Transformer backbone.

The architecture of Prot2RNA closely follows the Transformer encoder design introduced in Ri-NALMo (Penić et al., 2025), with some modifications. Specifically, we replace LayerNorm with RMSNorm (Zhang & Sennrich, 2019), which reduces computational overhead and improves efficiency. The innovation lies not in the backbone itself but in how the diffusion objective is applied

to aligned biological modalities, enabling iterative, context-aware codon generation rather than conventional masked-token prediction.

**Alphabet and Tokenization**   The model employs a shared vocabulary encompassing 21 amino acid tokens and 64 RNA codon tokens, allowing it to process protein and coding sequences within a unified representational space. Modality-specific special tokens denote sequence boundaries: [BOS_P] and [EOS_P] for proteins, [BOS_R] and [EOS_R] for RNAs, indicating the beginning and end of each sequence, respectively. Common tokens, such as [MASK] and [PAD], are used across both modalities. To preserve alignment between amino acids and codons, protein sequences are always terminated with the "$\star$" symbol, corresponding to the stop codon in RNA.

## 3.2 Data Preprocessing

We curated $141,193$ human protein–mRNA pairs by extracting protein sequences and their corresponding CDS from NCBI (Sayers et al., 2025) and GENCODE (Mudge et al., 2025). Only high-quality CDSs were retained, defined as those containing valid start and stop codons, having sequence length divisible by three, and being unique after removal of duplicates.

To enable generalization to unseen proteins, we constructed a test set of highly expressed sequences dissimilar to training examples. Expression levels were derived from the Human Protein Atlas (Uhlén et al., 2015; The Human Protein Atlas, 2025). We clustered all proteins at $80\%$ sequence identity and selected clusters containing at least one highly expressed pair. We chose the $3,000$ smallest such clusters, balancing the need for dissimilarity in the test set with preserving data for training. After filtering for protein length, the final test set consisted of $2,912$ protein-CDS pairs.

The remaining data were clustered at $90\%$ sequence identity and split 90:10 into training and validation sets. This higher identity threshold retained more sequences while maintaining separation between clusters. More details for the entire data preprocessing pipeline are provided in Appendix Section A.

## 3.3 Training

Prot2RNA is trained in two stages following a masked diffusion language modeling objective.

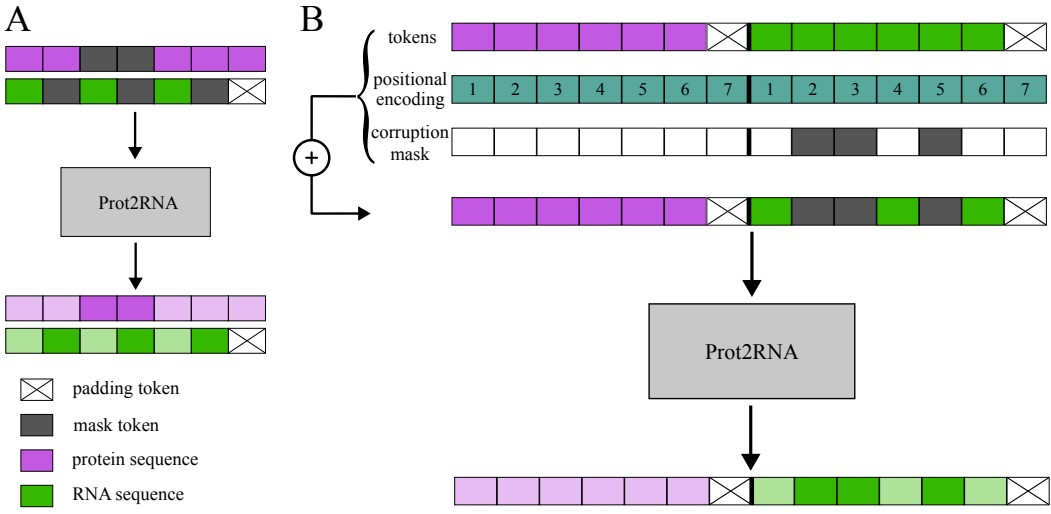

Figure 1: Two training stages of Prot2RNA. **A.** Pretraining stage: protein and RNA sequences are used as independent inputs, each masked with a different masking ratio $t \sim \mathcal{U}(0,1)$. The model is pretrained using masked diffusion modeling. **B.** Finetuning stage: the protein sequence serves as a prompt and remains unmasked, while only the RNA is masked. To reinforce the correspondence between protein and RNA, we apply shared positional encodings across the two segments.

**Pretraining Stage** In the first stage, the model is trained on individual protein and mRNA coding sequences. During the forward diffusion process, each token in a sequence is independently masked with a probability sampled from a uniform distribution $\mathcal{U}(0, 1)$. This generates a partially masked sequence $x_t$, for $t \sim \mathcal{U}(0, 1)$, where each token is masked with probability $t$. Prot2RNA is a parametric model, denoted as $p_\theta(\cdot|x_t)$, where $\theta$ represents the model parameters. It takes the partially masked sequence $x_t$ as input and simultaneously predicts all masked tokens.

The training objective has a simple form: it is the weighted average of masked language modeling losses, defined as:

$$\mathcal{L}(\theta) = -\mathbb{E}_{t,x_0,x_t} \left[ \frac{1}{t \cdot L} \sum_{i \in \mathcal{M}} \log p_\theta(x_0^i|x_t) \right], \tag{1}$$

where $x_0^i$ is the $i$-th token in a sequence $x_0$ of length $L$ sampled from the training set, $x_t$ is a corrupted version of $x_0$ sampled from the noising process and $\mathcal{M}$ is the index set of masked tokens.

Due to computational resources, we fix the model's context window to $1,024$ (including `BOS` and `EOS`). Because of this, sequences longer than this limit are randomly cropped to $1,022$ tokens in every epoch, following the strategy of (Rives et al., 2021). This stage encourages the model to learn general protein and codon-level statistics in a modality-agnostic manner.

**Finetuning Stage** The second stage finetunes the model on paired protein-RNA examples. The protein sequence $p_0$ is treated as the input prompt and remains fully visible (i.e., unmasked), while the corresponding RNA coding sequence $r_0$ is corrupted and serves as the target for generation. The masking ratio $t$ is again sampled uniformly between 0 and 1. We feed Prot2RNA with both the protein $p_0$ and the masked RNA $r_t$ to compute the loss for the finetuning stage:

$$\mathcal{L}_{FT}(\theta) = -\mathbb{E}_{t,p_0,r_0,x_t} \left[ \frac{1}{t \cdot L_r} \sum_{i \in \mathcal{M}} \log p_\theta(r_0^i|p_0, r_t) \right], \tag{2}$$

where $L_r$ denotes the RNA sequence length.

To ensure both sequences fit within the model's $2,048$-token context window, we restrict training to protein-CDS pairs shorter than $1,022$ amino acids and codons, respectively. During batching, protein sequences are first padded to the maximum protein length in the batch, and coding sequences are appended and then padded to the maximum CDS length. Since each amino acid aligns with a codon, this representation allows deterministic segmentation of the input into protein and RNA regions. Positional encodings are computed for the protein segment using RoPE for the protein portion (positions 1 to $L_p$), and these same encodings are reused for the RNA portion (positions 1 to $L_r$), such that codon $i$ and amino acid $i$ receive the same positional information. This modality-aligned encoding strategy, inspired by (Wang et al., 2025), promotes tight coupling between the two representations and enables the model to attend bidirectionally across the entire input while maintaining alignment-aware structure. We found that this shared encoding scheme eliminated mistakes in amino acid preservation and additionally improved codon-level accuracy (see Appendix Subsection E.2).

In addition to training on all paired examples, we further finetuned the same model on a subset of $25,274$ highly expressed protein-RNA pairs from the training set. This additional stage biases the model toward codon patterns associated with elevated expression levels. We refer to this variant as Prot2RNA_FT2, and compare its performance against the base model to evaluate whether further specialization improves generation quality.

In all training stages, the model is optimized using a standard cross-entropy loss over masked tokens. Only positions corresponding to masked tokens contribute to the loss; unmasked and `[PAD]` tokens are ignored. During pretraining, the loss is computed jointly over both protein and RNA sequences within each batch. Unlike the iterative denoising used at generation time, training is performed in a single denoising step per sequence, following the masked diffusion modeling paradigm.

Complete training setup details are provided in Appendix Section B.

### 3.4 SEQUENCE GENERATION

To generate the mRNA coding sequence from a given target protein, Prot2RNA uses a reverse diffusion process to iteratively denoise a fully masked RNA sequence. Starting from an initial sequence

composed entirely of masked tokens $r_1$, the model progressively refines the sequence over multiple timesteps until the final sequence $r_0$ is generated. The number of denoising iterations and the masking schedule, defined by the function $\gamma(\cdot)$ that determines the masking ratio $t$ at each step and was adapted from (Chang et al., 2022), were treated as tunable hyperparameters that influence the quality and biological properties of the generated mRNA coding sequence. We evaluated several masking uniform and nonuniform schedules and different numbers of denoising steps, as detailed in Section 4.4 and Appendix Subsection F.

At each intermediate denoising step, the model takes as input the target protein sequence $p_0$ and a partially masked mRNA coding sequence $r_t$ and predicts all masked tokens in parallel. Based on the model's prediction confidences, a subset of tokens—those with the lowest confidence—is selected and remasked. The fraction of tokens to be remasked at each step is determined by the masking scheduling function $\gamma$. For clarity and reproducibility, we include a pseudocode description of the full generation algorithm in the Appendix Algorithm 1.

## 4 RESULTS & DISCUSSION

To evaluate Prot2RNA, we compared its performance to several commonly used and novel methods, including CodonBERT, CodonTransformer (C-Transformer), LinearDesign, and Trias. As reference points, we included wild-type sequences and a naïve CAI-maximized (CAI-max) baseline. Performance was assessed using established metrics, including codon-level accuracy, CAI, GC content, and MinMax profiles, providing a comprehensive view of sequence fidelity, codon usage bias, and composition-level similarity to wild-type sequences More detailed definitions of these metrics are provided in Appendix Section C..

### 4.1 EVALUATION OF CODON-LEVEL ACCURACY

We assess how well each model preserves natural codon usage by measuring codon-level accuracy (the proportion of codons in the generated sequence that exactly match those in the wild-type). This metric is especially relevant since our test set consists of highly expressed native sequences, and we hypothesize that closely mimicking their codon patterns could support similarly high expression.

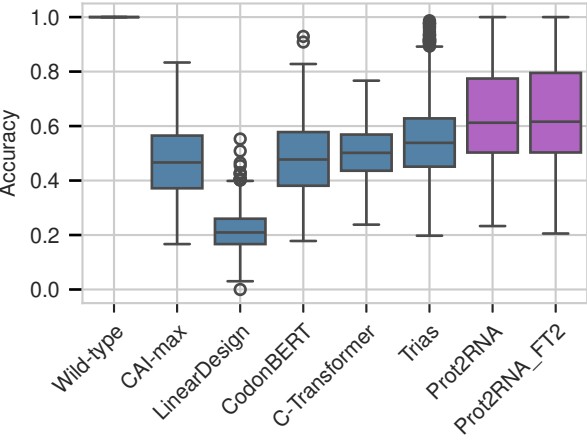

Figure 2: Codon-level accuracy of sequences generated by different methods, evaluated against wild-type references.

Figure 2 summarizes codon-level accuracy across evaluated models. By design, wild-type sequences achieve a perfect accuracy of $1.0$, since they are compared against themselves. Among all models, Prot2RNA_FT2 achieves the highest median codon-level accuracy at $0.616$, outperforming all other methods, including the base Prot2RNA model ($0.612$). In contrast, LinearDesign yields the lowest median codon-level accuracy ($0.209$). While it jointly optimizes for codon usage and structural stability, this objective often favors codon substitutions that deviate from natural codon patterns, leading to low overlap with wild-type sequences. Trias achieves intermediate accuracy ($0.538$),

possibly reflecting a trade-off between its large, multi-species training corpus and generalization to human sequences. CodonTransformer (0.502) and CodonBERT (0.477), despite being pretrained or fine-tuned with frequency-based objectives, perform only marginally better than the naive CAI-maximized baseline (0.466). This indicates that codon frequency-driven methods provide little advantage over simple frequency maximization, and fail to capture the nuanced codon usage of highly expressed human sequences.

Lastly, amino acid accuracy remains high for all models (median amino acid sequence fidelity values are 1.00), but only Prot2RNA models combine this with close codon-level matching, highlighting their potential to mimic biologically relevant codon choices.

## 4.2 BIOLOGICAL PROPERTIES OF GENERATED SEQUENCES

Next, we evaluated the CAI of generated sequences to assess how well each model aligns with the codon preferences typical of highly expressed genes (see Figure 3A).

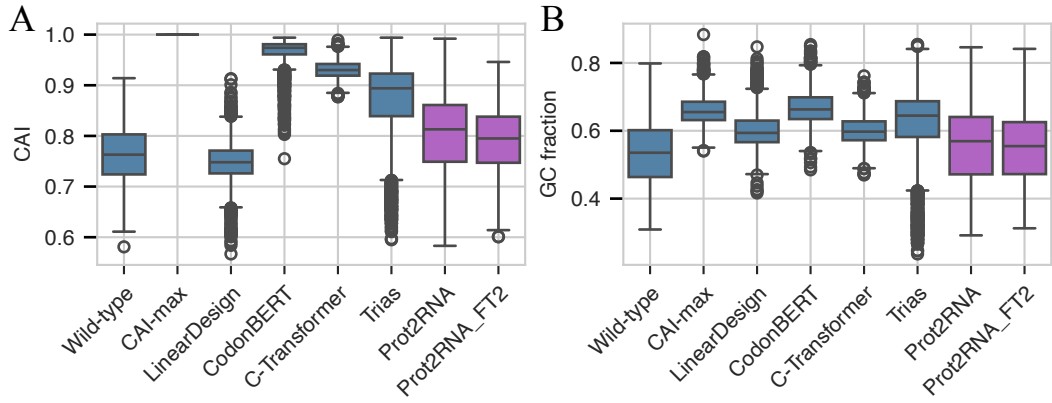

Figure 3: Biological properties of sequences generated by different methods evaluated against wild-type references. A. Codon Adaptation Index (CAI). B. GC content.

As expected, the CAI-maximized baseline achieves a perfect CAI of 1.0 for all sequences, since it deterministically selects the most frequent synonymous codon for each amino acid. In contrast, wild-type sequences achieve a median CAI of 0.763, reflecting natural codon usage where high expression is not driven solely by codon bias, but a range of other factors, as previously shown (Vogel et al., 2010). Among other models, CodonBERT yields the highest CAI (0.974), consistent with its pretraining on JCAT-optimized sequences that maximize codon usage bias (Grote et al., 2005). Prot2RNA and Prot2RNA_FT2, on the other hand, produce sequences with more moderate CAI values (0.813 and 0.795, respectively), closely resembling the wild-type distribution. Interestingly, Prot2RNA_FT2 shows a slight drop in CAI despite improved codon-level accuracy, suggesting that additional training on high expression sequences helps avoid over-optimization and better mimics biologically plausible codon usage. Other models, such as CodonTransformer (0.930) and Trias (0.894), fall between these extremes, balancing codon bias and fidelity to natural usage. LinearDesign achieves a median CAI value (0.748), closely matching that of wild-type sequences, consistent with its joint objective that focuses on codon usage and structural stability.

We further calculated the Guanine-Cytosine (GC) content of the generated sequences (Figure 3B), known to influence mRNA stability and translational efficiency critically. GC base pairs form three hydrogen bonds, compared to two in adenine–uracil pairs, making them thermodynamically more stable. While moderate GC composition can improve mRNA half-life by preventing degradation, excessive stability may hinder translation by inducing ribosomal stalling. Prior studies in human cell lines suggest that optimal GC content for efficient expression lies between 50–60% (Hia et al., 2019; Presnyak et al., 2015b).

As shown in Figure 3B, wild-type sequences show a median GC content of 53.5%. Prot2RNA and Prot2RNA_FT2 yield similar distributions, with median GC values of 56.9% and 55.5%, respec-

tively, closely mirroring the wild-type profile, and indicating the ability to preserve biologically relevant composition. In contrast, CAI-maximized (65.5%), CodonBERT (66.3%), and Trias (64.5%) exhibit elevated GC levels. This is likely due to strong optimization pressures favoring frequently used, GC-rich codons, which potentially improve stability, but at the cost of translational efficiency. CodonTransformer (59.7%) and LinearDesign (59.4%) also fall near the upper end of the optimal GC range, indicating a moderate bias toward GC-rich codons, while remaining more balanced than frequency-maximizing methods.

Additional analysis of GC3 content (the proportion of codons ending in G or C) supports the conclusion that Prot2RNA preserves biologically realistic sequence properties (see Appendix Figure 9). Wild-type sequences show a median GC3 of 60.2%, and Prot2RNA_FT2 closely matches this (66.5%). In contrast, CAI-maximized (99.5%) and CodonBERT (97.7%) outputs are heavily skewed toward GC-rich codons, reflecting aggressive codon frequency optimization that may impair translational efficiency.

To additionally evaluate whether the generated coding sequences are semantically similar to natural ones, we propose a novel distance measure between two sets of coding sequences. Similar to the Fréchet Inception Distance (FID) (Heusel et al., 2017), an established performance metric for comparing image generation methods, we calculate the Fréchet distance between Gaussian-approximated distributions of their [CLS] embeddings from a fixed encoder-only CodonBERT model pretrained on natural coding sequences (Li et al., 2024). This encoder is used solely as a feature extractor and is distinct from the CodonBERT generative baseline evaluated elsewhere in this study. Our metric has an advantage over previous metrics in that it can detect if generated coding sequences have global semantic properties as real ones. Prot2RNA achieved the lowest divergence from wild-type sequences on the held-out test set (1.39), while baseline methods showed substantially higher distances (CAI-max 52.3, CodonBERT 28.0, CodonTransformer 11.0, LinearDesign 8.5, Trias 14.2). As a complementary visualization, we projected CodonBERT embeddings into the first two principal components (Appendix Figure 11), where Prot2RNA_FT2 clustered closer to wild-type sequences than alternative methods, consistent with the quantitative results. Additional analysis of Fréchet distance behavior is provided in Appendix Section G.

### 4.3 MinMax Profiles

To examine local codon usage patterns beyond global sequence-level metrics, we analyze the MinMax profiles of each sequence. A MinMax profile captures position-wise codon optimality by scaling the relative adaptiveness of each codon within its synonymous group to a range from $-100\%$ (least adapted) to $+100\%$ (most adapted). This normalization offers a position-specific view of codon bias, enabling assessment of how different models modulate codon choice across transcript length (Rodriguez et al., 2018).

Figure 4A presents MinMax profiles for one example transcript across five representative tools. For each model, the generated sequence (purple or blue) was plotted alongside the corresponding wild-type sequence (gray). Prot2RNA displays smooth and coherent MinMax trends that closely mirror the wild-type profile. In contrast, Trias and CodonBERT exhibit flatter or highly saturated profiles, often clustering near the $+100\%$ boundary, indicating overreliance on the most frequent codons. CodonTransformer follows a similar pattern, while LinearDesign shows less consistent behavior, likely reflecting its structure-driven optimization objective. In contrast to other deep learning models, these profiles suggest that Prot2RNA is the only one that effectively preserves the natural codon usage landscape.

To quantify how closely each generated sequence mirrors the codon usage profile of its corresponding wild-type sequence, we computed the mean Euclidean distance between their respective Min-Max profiles across the test set (Figure 4B). Lower distances indicate stronger alignment with natural codon usage patterns. Prot2RNA achieves the lowest median distance among all models, reinforcing its ability to preserve fine-grained positional codon trends found in highly expressed transcripts. In contrast, CodonBERT, CodonTransformer, and CAI-maximized sequences deviate substantially from wild-type profiles, likely due to their strong bias toward codon frequency maximization. LinearDesign, which jointly optimizes codon usage and secondary structure, also yields relatively low distances. This suggests that its objective leads to codon distributions that, while not trained on wild-type data, still resemble natural profiles to some extent. Overall, these findings suggest that

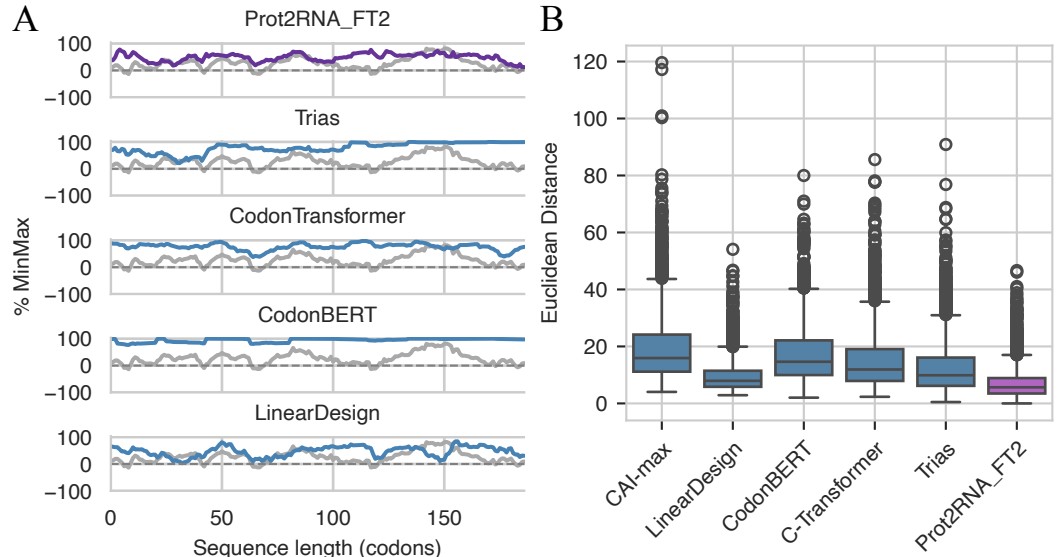

Figure 4: Comparison of local codon usage profiles between generated and wild-type sequences. A. MinMax profile for the coding sequence NM_001144932.3. The wild-type sequence is shown in gray, and the generated sequences are shown in purple and blue. B. Distribution of mean Euclidean distances between generated and wild-type MinMax profiles.

Prot2RNA captures not only global codon preferences but also local usage patterns, which may be critical to maintaining regulatory elements and achieving high expression efficiency.

## 4.4 DESIGN CHOICE ANALYSIS

**Scaling Laws** We evaluated three Prot2RNA versions of increasing size: nano (8.5M parameters), micro (33.5M), and mega (153M). As shown in Appendix Subsection E.1, codon-level accuracy improves consistently with model size, suggesting that larger models are better able to capture codon usage preferences from training data. At the same time, larger models produce sequences with lower CAI and more balanced GC content, which more closely resemble the characteristics of highly expressed wild-type sequences. Given our objective to mimic the codon patterns of naturally highly expressed sequences, we selected the mega model for all primary experiments, balancing expressiveness with biological realism.

**Decoding Strategies** As shown in Appendix Section F, we find that among the various masking schedules, the cosine function consistently achieves the highest codon-level accuracy across decoding configurations. In this scheme, the process begins with all tokens masked, and at each subsequent iteration an increasingly larger portion of tokens becomes unmasked and updated. Thus, early steps perform coarse global predictions under high noise (few visible tokens), while later steps refine many tokens simultaneously as more context is revealed. This aligns with findings from previous work (Chang et al., 2022), further supporting its suitability for denoising-based generation. Based on this, we adopt the cosine masking schedule for all remaining experiments. Next, we evaluated the effect of varying the number of decoding iterations, using the cosine schedule. We observe that accuracy improves with more iterations up to a point, with diminishing or no gains beyond four steps. In particular, using four iterations with cosine masking yields the highest mean and median codon-level accuracy among all tested configurations. We therefore select this setting, using a cosine schedule with four iterations, as the default decoding strategy for Prot2RNA and its finetuned variant Prot2RNA_FT2.

## 5 CONCLUSIONS

Prot2RNA accurately replicates codon usage patterns found in highly expressed human genes, outperforming both traditional and deep learning baselines across key metrics. It achieves the highest codon-level accuracy while maintaining CAI, GC and GC3 content distributions close to wild-type levels, striking a biologically relevant balance between expression optimization and sequence fidelity. Unlike frequency-maximizing models, Prot2RNA captures context-dependent synonymous codon biases, reflecting position- and motif-specific patterns of codon choice found in natural human transcripts and resulting in moderate CAI (0.79) and GC3% (69%) values. Fine-grained MinMax analyses confirm its ability to preserve local codon usage patterns. Importantly, these results are achieved without handcrafted rules or post-hoc adjustments, underscoring the method's generalizability and scalability for mRNA design.

Despite these strengths, Prot2RNA has several limitations. It is trained solely on sequence data, without direct integration of experimental gene expression measurements, and its outputs have not yet been validated in in vivo or in vitro expression assays. This represents the most important next step for establishing the functional impact of Prot2RNA-designed sequences. The model also focuses only on coding regions, omitting UTRs, RNA secondary structure, and translation dynamics, all of which influence mRNA stability, localization, and translation efficiency. Architecturally, it inherits the computational demands of large transformer models and does not incorporate biological priors such as folding constraints or ribosomal behavior, which may limit interpretability and generalizability.

Looking ahead, experimental validation will be essential to assess whether Prot2RNA-designed sequences yield improved protein expression *in vivo*. Future work could extend the model to incorporate UTRs, structural feedback during generation, or training on ribosome profiling and mRNA half-life data. These additions could enhance the model's biological fidelity and broaden its utility in both basic research and therapeutic mRNA design.

### REPRODUCIBILITY STATEMENT

We fixed the random seed to 42 across PyTorch, NumPy, and Python's random module to ensure deterministic behavior during model training and sequence generation, and report results from a single run per experiment. Full training setup details are provided in Appendix Section B.

Our code, dataset, and model weights for all Prot2RNA variants are available at `https://figshare.com/s/414056ccf253acb31a4a`. For baseline models (CodonBERT, Trias, LinearDesign, CodonTransformer), we used the official implementations, with further details provided in Appendix Section H.

### ETHICS STATEMENT

This work uses only publicly available data and does not involve human subjects, sensitive personal information, or experiments with potential biohazards. We do not foresee direct negative societal or ethical impacts.

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

## A    DATA PREPROCESSING

We curated a dataset of $141,193$ human protein–mRNA pairs by extracting protein sequences and their corresponding mRNA coding sequences from NCBI RefSeq (Sayers et al., 2025) and GEN-CODE (Mudge et al., 2025) (version 48, file *gencode.v48.pc_translations.fa.gz*). Only CDS regions were retained for each mRNA transcript. Only coding sequences with a length divisible by three, starting with a start codon, ending with a stop codon, and only having a single stop codon were chosen. To ensure diversity and prevent model overfitting, we removed duplicate CDS entries, keeping only unique coding sequences. Our objective was to train a model that can generate highly expressed mRNA sequences for novel protein inputs, i.e. proteins not encountered during training. To facilitate this, we designed a test set composed exclusively of highly expressed examples that share low sequence similarity with the training set.

Highly expressed mRNAs were identified using median Transcripts Per Million (TPM) values from the Human Protein Atlas (Uhlén et al., 2015; The Human Protein Atlas, 2025) (version 23.0, file *transcript_rna_tissue.tsv*). A coding sequence was labeled as "highly expressed" if its median TPM exceeded 5 in at least one tissue. Since HPA provides only transcript identifiers, we retrieved the corresponding sequences from GENCODE, which uses compatible IDs. To ensure label consistency, we excluded cases where the same CDS appeared under multiple transcript IDs with conflicting TPM values. Specifically, we retained only those CDSs for which all associated transcripts consistently showed median TPM above 5. This yielded $25,274$ high expression protein–CDS pairs with unique coding sequence.

To construct a suitable test set, we first clustered all protein sequences using MMseqs2 (Steinegger & Söding, 2017) at 80% sequence identity threshold. From these clusters, we identified those containing at least one highly expressed sequence. To ensure sequence novelty and diversity, we selected the $3,000$ smallest clusters, i.e., those with the fewest members, thereby minimizing similarity to the training data. To comply with model input length constraints, we further filtered the sequences by retaining only those with protein lengths under $1,022$ amino acids. This resulted in a final test set comprising $2,912$ unique protein–CDS pairs.

After finalizing the test set, we removed all protein-CDS pairs belonging to their MMseqs2-defined clusters from the remaining dataset to prevent data leakage. We then reclustered the remaining protein sequences at 90% sequence identity to reduce redundancy and enforce lower similarity between splits. These clusters were then partitioned into training and validation sets using a 90:10 ratio, resulting in distinct sets with minimal sequence overlap.

To characterize the overall sequence-length landscape of the dataset, we examined the CDS length distributions of all curated sequences, all sequences that fall within the model's $1,022$-codon context window, and the sequences in the final test split. The distributions show broadly similar shapes and span typical human CDS lengths (Figure 5). The medians for the full set, the in-context subset, and the test set are 488, 400, and 206 codons, respectively, indicating that the context-window constraint mainly removes very long transcripts while preserving the majority of natural variation.

## B    TRAINING SETUP

During the pretraining stage, we adopted the cosine annealing learning rate schedule with a linear warm-up. Namely, we linearly increased the learning rate from 0 to $1 \times 10^{-4}$ over the first $2,000$ training steps. Subsequently, we annealed the learning rate to $1 \times 10^{-5}$ during the next $50,000$ training steps. We used the AdamW optimizer (Loshchilov & Hutter, 2017) with a weight decay of $0.01$. We pretrained Prot2RNA using $4$ A100 GPUs of 80 GB memory with a batch size of 384 per GPU.

In the first finetuning stage, we again used linear warm-up and cosine annealing learning rate schedulers. We linearly increased the learning rate from 0 to $3 \times 10^{-5}$ and then we decayed the learning rate to $8 \times 10^{-6}$ for $50,000$ steps. In the second finetuning stage, we used only a cosine annealing learning rate scheduler and we decayed the learning rate from $2 \times 10^{-5}$ to $1 \times 10^{-5}$ for 50 epochs. During both finetuning stages, we utilized the AdamW optimizer with a weight decay of $0.01$. Furthermore, we used 2 A100 GPUs of 80 GB memory with a batch size of 128 per GPU.

## C  SEQUENCE GENERATION

To generate the sequence, we used the reverse denoising procedure. We provide the pseudocode in Algorithm 1. We tried different iterative denoising schedules employing the mask scheduling function $\gamma(\cdot)$ and various numbers of steps. Different mask scheduling functions are provided in Appendix Section F.

---

**Algorithm 1** Prot2RNA Iterative mRNA Coding Sequence Generation

---

**Require:** protein prompt $p_0$, denoising steps $N$, masking scheduling function $\gamma$
 1: Set $r_t$ as a fully masked sequence: $r_t \leftarrow r_1$
 2: All the positions are masked in the beginning: $\mathcal{M} \leftarrow \{1, \ldots, L_r\}$
 3: **for** $n \leftarrow 1$ **to** $N$ **do**
 4:     **for** $i \leftarrow 1$ **to** $L_r$ **do**
 5:         **if** $i \in \mathcal{M}$ **then**
 6:             $r_0^i = \arg\max_{r_0^i} p_\theta(r_0^i|p_0, r_t)$
 7:             $c^i = p_\theta(r_0^i|p_0, r_t)$
 8:         **else**
 9:             $r_0^i = r_t^i$
10:             $c^i = 1$
11:         **end if**
12:     **end for**
13:     $n_m \leftarrow \lfloor \gamma(\frac{n}{N}) * L_r \rfloor$         # the number of tokens masked in the next step
14:     $\mathcal{M} \leftarrow \emptyset$
15:     **for** $i \leftarrow 1$ **to** $L_r$ **do**
16:         **if** $c^i \in$ Lowest $n_m$
17:             $\mathcal{M} \leftarrow \mathcal{M} \cup \{i\}$         # the $n_m$ positions with the lowest confidence are remasked
18:         **end if**
19:     **end for**
20:     $r_t \leftarrow r_0$
21: **end for**
22: **return** $r_0$

---

## D  EVALUATION METRICS

This section provides detailed definitions and formulas for the metrics used to characterize generated coding sequences.

### D.1  CODON ADAPTATION INDEX (CAI)

The Codon Adaptation Index (CAI) measures how closely a coding sequence uses codons preferred in a reference set of highly expressed human genes. For a sequence with codons $c_1, \ldots, c_L$, CAI is defined as

$$\text{CAI} = \left( \prod_{i=1}^{L} w_{c_i} \right)^{1/L}, \tag{3}$$

where $L$ is the number of codons in the sequence and $w_c \in [0, 1]$ is the relative adaptiveness weight of codon $c$,

$$w_c = \frac{f_c}{\max_{c' \in \mathcal{S}(a)} f_{c'}}, \tag{4}$$

with $f_c$ denoting the empirical frequency of codon $c$ in a high-expression reference set and $\mathcal{S}(a)$ the set of synonymous codons for amino acid $a$. CAI is a compositional metric and is used here only to ensure that sequences remain within typical human codon-usage ranges.

## D.2 GC CONTENT AND GC3 CONTENT

For an RNA sequence of length $N$, GC content is

$$\text{GC\%} = \frac{\#(G) + \#(C)}{N}. \tag{5}$$

GC content at third codon positions (GC3%) is

$$\text{GC3\%} = \frac{\#(G_3) + \#(C_3)}{L}, \tag{6}$$

where $L$ is the number of codons and $G_3, C_3$ denote nucleotides at the third position of each codon. These compositional statistics help verify that generated sequences fall within biologically plausible ranges observed in human CDSs.

## D.3 MINMAX PROFILES

The MinMax metric (Rodriguez et al., 2018) provides a codon-level measure of optimality based on relative adaptiveness within each synonymous group. For each codon $c$ encoding amino acid $a$, let $f_c$ denote its empirical frequency in a reference set of highly expressed human genes. The relative adaptiveness is

$$w_c = \frac{f_c}{\max_{c' \in \mathcal{S}(a)} f_{c'}}, \tag{7}$$

where $\mathcal{S}(a)$ is the set of synonymous codons for amino acid $a$. The MinMax score rescales $w_c$ to the range $[-100, +100]$:

$$\text{MinMax}(c) = 200 \cdot \frac{w_c - \min_{c' \in \mathcal{S}(a)} w_{c'}}{\max_{c' \in \mathcal{S}(a)} w_{c'}} - 100. \tag{8}$$

A MinMax profile for a coding sequence $r = (c_1, \ldots, c_L)$ is the vector of scores

$$\big(\text{MinMax}(c_1), \ldots, \text{MinMax}(c_L)\big),$$

where $L$ is the number of codons in the sequence.

To summarize deviation from reference human codon optimality, we compute **Euclidean MinMax distance**:

$$D_{\text{Euc-MinMax}}(r) = \sqrt{\sum_{i=1}^{L} \big(\text{MinMax}(c_i) - \text{MinMax}^{\text{ref}}(c_i)\big)^2}. \tag{9}$$

Here, $\text{MinMax}^{\text{ref}}(c_i)$ denotes the MinMax score of the ground-truth test codon at position $i$. These metrics allow us to quantify how closely a generated sequence reproduces codon optimality patterns present in its ground-truth counterpart, which are in our case highly expressed human sequences.

# E ABLATION STUDY

## E.1 MODEL SIZE COMPARISON

We compared three Prot2RNA versions of increasing size: nano (8.5M parameters), micro (33.5M), and mega (153M). Figure 6 shows how these models perform in terms of codon-level accuracy (A), CAI (B), and GC content (C). D part of the figure shows architectural hyperparameters for each model size. We see that codon accuracy improves consistently with model size, suggesting that larger models are better able to capture codon usage preferences from training data. Given our objective to mimic the codon patterns of naturally highly expressed sequences, we selected the mega model for all primary experiments, balancing expressiveness with biological realism. We therefore proceeded with the mega model for the second stage of finetuning on highly expressed sequences (Prot2RNA_mega_FT2).

## E.2 POSITIONAL ENCODINGS

To assess the impact of modality-aligned positional encoding, we compared the Prot2RNA_nano model with our modified RoPE (shared across protein and RNA segments) against a baseline with standard RoPE, where rotary encodings are computed over the entire concatenated sequence (protein + RNA). As shown in Appendix Figure 7, the modality-aligned version achieved higher codon-level accuracy (median 0.512 vs. 0.490). At the amino acid level, both variants reached a median accuracy of 1.0, but the standard RoPE occasionally produced outliers ($0.9995 \pm 0.011$), whereas our modified RoPE eliminated such errors ($1.0000 \pm 0.0001$). This robustness was important: even at the nano scale, the model learns to select codons that consistently preserve amino acids, without relying on handcrafted constraints that force synonymous codon choice. Based on this improvement in accuracy, we adopted the modality-aligned RoPE for all larger models in subsequent experiments.

## F ITERATIVE MASK SCHEDULING

In Figure 8, we provide different mask scheduling functions $\gamma(\cdot)$ tested for the reverse denoising process.

Appendix Table 1 reports the median and mean codon-level accuracy (± standard deviation) for sequences generated using Prot2RNA_FT2 with different mask scheduling functions and 4 denoising steps. Since the cosine function yielded the best performance, we further explored how the number of denoising steps influences accuracy using this schedule.

Table 1: **Codon-level accuracy for different masking schedule functions used during sequence generation with Prot2RNA_FT2 and 4 denoising steps.**

| Mask scheduling function | Median Accuracy | Mean ± Std |
|---|---|---|
| Cubic | 0.6154 | 0.6528 ± 0.1840 |
| Square | 0.6154 | 0.6520 ± 0.1835 |
| **Cosine** | **0.6154** | **0.6520 ± 0.1833** |
| Linear | 0.6151 | 0.6517 ± 0.1828 |
| Square root | 0.6145 | 0.6518 ± 0.1820 |

Appendix Table 2 reports codon-level accuracy for different numbers of denoising steps used during sequence generation with the Prot2RNA_FT2 model and the cosine masking schedule function. Since 4 steps yielded the best results, we use the cosine schedule with 4 denoising steps for all other Prot2RNA evaluations.

Table 2: **Codon-level accuracy for different numbers of denoising steps used during sequence generation with Prot2RNA_FT2 and cosine masking schedule function.**

| Number of denoising steps | Median Accuracy | Mean ± Std |
|---|---|---|
| 1 step | 0.6078 | 0.6511 ± 0.1853 |
| 2 steps | 0.6123 | 0.6533 ± 0.1847 |
| **4 steps** | **0.6161** | **0.6536 ± 0.1838** |
| 8 steps | 0.6154 | 0.6520 ± 0.1833 |
| 12 steps | 0.6147 | 0.6518 ± 0.1827 |

## G ADDITIONAL RESULTS

Appendix Figure 9 shows the distribution of GC3 content, defined as the percentage of codons ending in G or C. Wild-type sequences display a broad and biologically typical GC3 distribution centered around 60.2%. In contrast, CAI-maximized, CodonBERT, and LinearDesign outputs are strongly biased toward GC-ending codons, with median values near or above 95%. Prot2RNA and Prot2RNA_FT2 generate sequences with more moderate GC3 values, closely matching the wild-type distribution, indicating a more balanced codon usage pattern.

## G.1 Fréchet Distance Evaluation

To quantify distributional similarity between natural and generated coding sequences, we compute the Fréchet distance between Gaussian-approximated distributions of CodonBERT [CLS] embeddings. Because the Fréchet estimator is sensitive to finite-sample noise in the empirical means and covariances, we examined how the metric behaves as a function of sample size.

**Finite-sample behavior.** We measured the Fréchet distance between two non-overlapping random subsets of natural coding sequences of varying sizes. Figure 10 shows the resulting curve. As expected, the distance decreases monotonically with sample size and stabilizes once both subsets contain several thousand sequences. The curve approaches a noise floor of roughly $0.5 \pm 0.01$ at $10,000$ samples per set, indicating that at this scale the estimator variance is negligible compared to the effect sizes observed between real and generated distributions. This provides a principled upper bound on the sampling noise inherent to our embedding space.

Because generating $10,000$ full-length coding sequences is computationally expensive for some baselines (in particular LinearDesign), the main paper reports all Fréchet distances using the $2,912$-sequence test set to ensure comparability across methods. To evaluate whether the ranking of models depends on sample size, we additionally generated $10,000$ sequences for all models except LinearDesign and recomputed the distances. These sequences were generated using proteins from the validation set, which our model never observed during training. Although absolute values decrease slightly due to reduced estimator variance, the ordering of methods remains unchanged, indicating that the test-set-scale evaluation used in Section 4.2 provides reliable relative comparisons. The results are presented in table 3

| Model | FD (10k) | FD (test-set) |
|---|---|---|
| Prot2RNA | 0.503 | 1.387 |
| LinearDesign | — | 8.456 |
| CodonTransformer | 11.20 | 11.029 |
| Trias | 11.85 | 14.24 |
| CodonBERT | 33.717 | 27.977 |
| CAI-max | 71.235 | 52.336 |

Table 3: Comparison of Fréchet distances for large-sample (10k) and test-set-sized evaluations.

A complementary visualization of CodonBERT [CLS] embeddings for the sequences from the test set projected into the first two principal components is shown in Figure 11.

## G.2 Comparison to commercially available codon optimization tools

To compare Prot2RNA with commercially used codon-optimization tools, we evaluated a subset of the test set for which commercial sequences could be feasibly obtained. Because some of widely used web-based services accept only small batch submissions and exhibit long turnaround times, we selected a representative subset of 200 proteins for which commercial optimization was practical.

To construct this subset, we first recorded the amino-acid lengths of all 2,912 proteins in the test set and stratified them into length bins (0–200, 200–400, 400–600, 600–800, and 800–1022 aa). We then sampled up to 40 proteins per bin to approximate the true test-set length distribution and ensure coverage across length regimes. These 200 proteins were provided as inputs to Prot2RNA, IDT (Integrated DNA Technologies, 2024), and Genewiz (Azenta Life Sciences, 2024), yielding one optimized coding sequence per method for each protein.

All comparative analyses (GC%, GC3%, CAI, and codon-level accuracy) were performed on this matched set of 200 protein–CDS outputs, enabling a controlled, sequence-aligned evaluation of Prot2RNA alongside commercial tools. The results for these metrics are illustrated in Figure 12.

## H    REPRODUCIBILITY DETAILS FOR OTHER MODELS

To calculate CAI, we used the `cai` utility and codon usage table "Ehuman.cut" from the EMBOSS software package (version 6.6.0) (Olson, 2002).

For CodonBERT, we removed the postprocessing step present in the original implementation. This step replaces invalid codons with predefined canonical codons to guarantee 100% amino acid preservation, stipulated by the `fix_AA_codon` lookup table. However, this may overwrite model-inferred codons, thereby distorting evaluation. To ensure fair comparison based solely on model output, we report results directly from the raw predictions.

For Trias, the model was given the flag `--species 'Homo Sapiens'` at inference, and the greedy decoding strategy was used.

For LinearDesign, the hyperparameter $\lambda$, which balances MFE and CAI terms in their objective function, was set to the default value of 0.

For CodonTransformer, we used the variant fine-tuned on human data, as released by the original authors by setting the `organism` parameter to 'Homo sapiens'. All other settings (e.g., attention and decoding parameters) were kept at their default values.

## STATEMENT ON LLM USAGE

We used ChatGPT (GPT-4/5) as a general-purpose writing assist tool, limited to text polishing and improving clarity of presentation. The model was not used for research ideation, analysis, coding, or experimental design. All scientific content and conclusions are solely the responsibility of the authors.

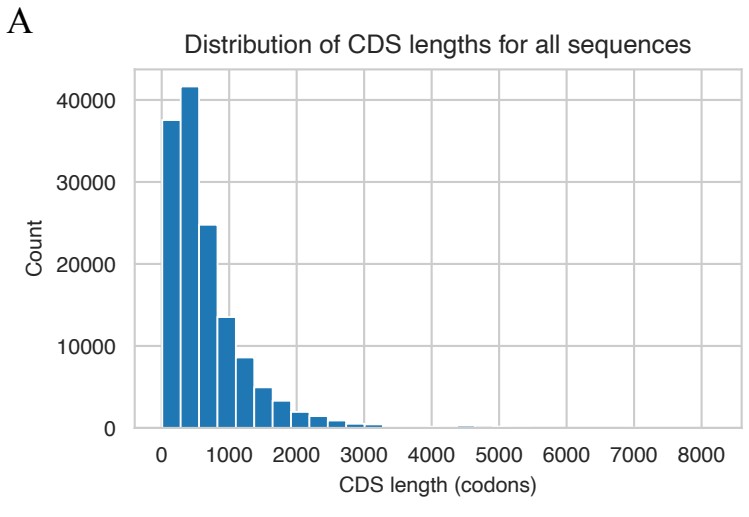

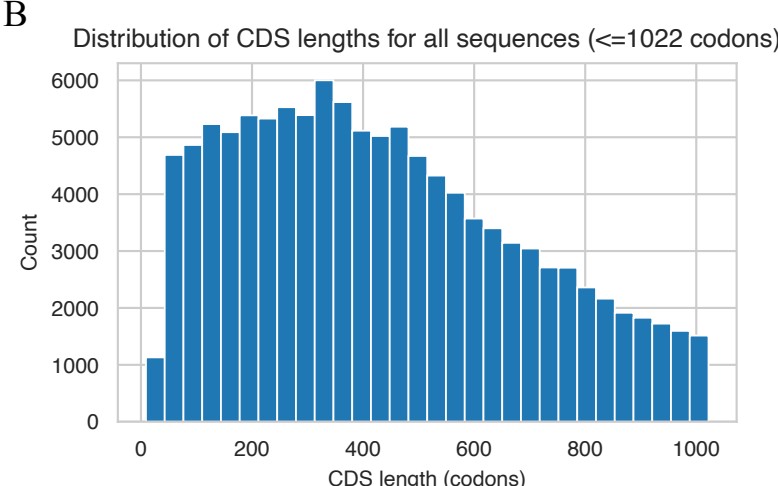

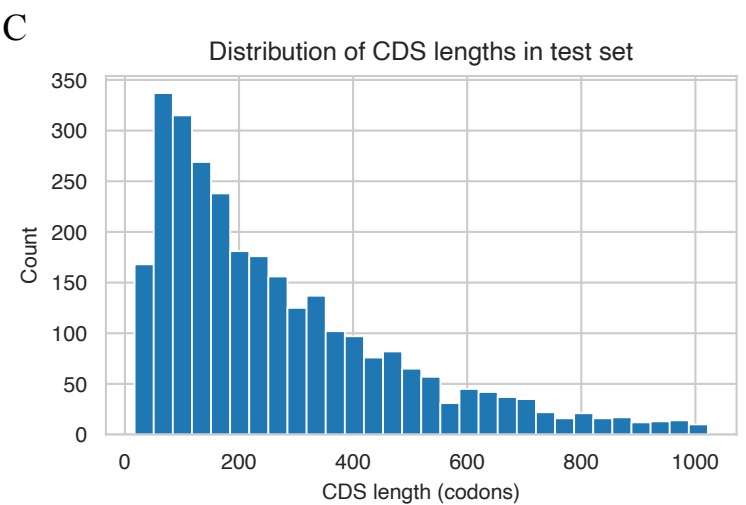

Figure 5: CDS length distributions. **A.** All human coding sequences in the dataset (median 488 codons). **B.** All coding sequences that fit within the model's 1,022-codon context window (median 400 codons). **C.** The 2,912 coding sequences in the final test set (median 206 codons).

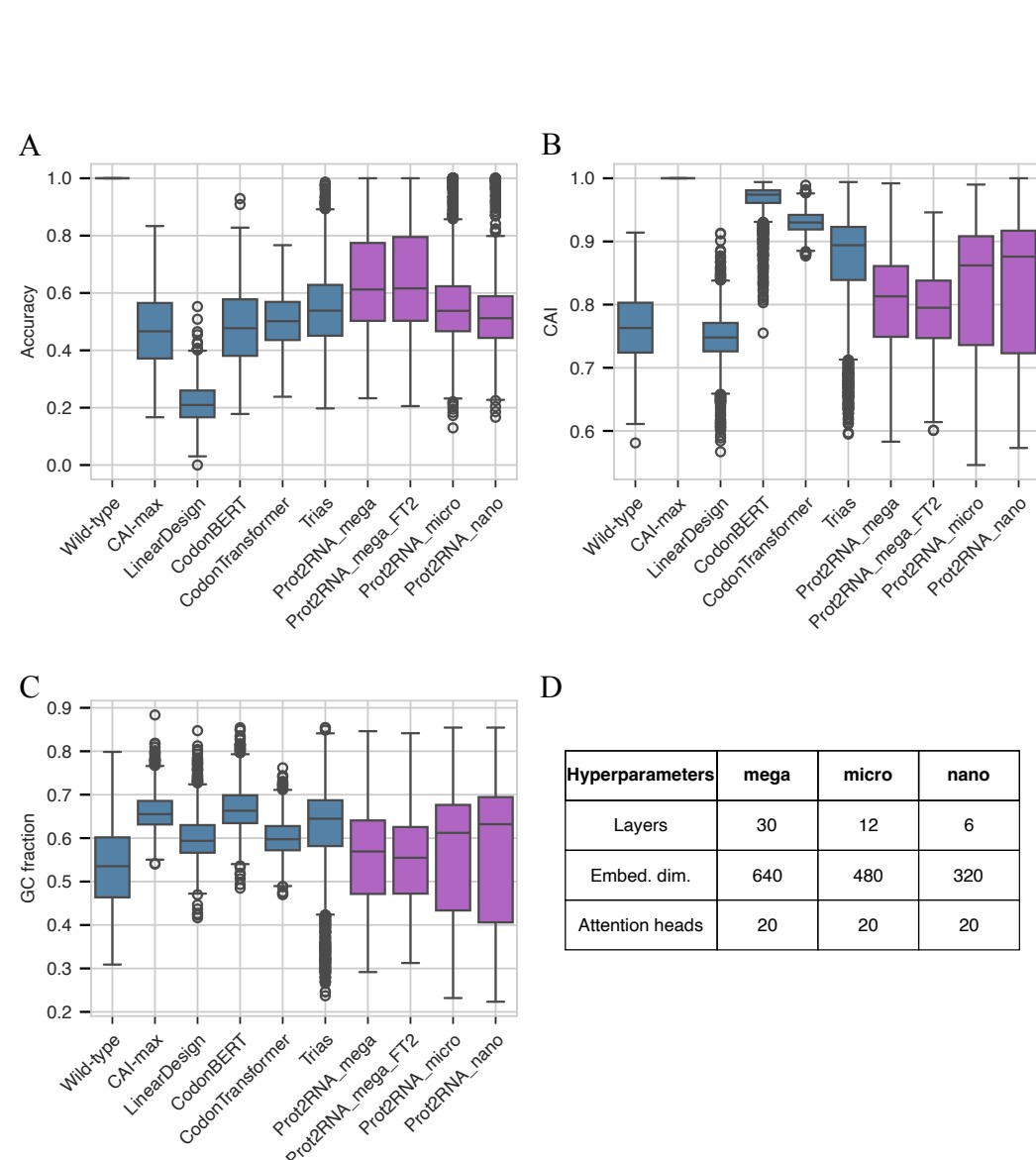

Figure 6: Performance comparison of different model sizes. All Prot2RNA models use a fixed cosine masking schedule and 4 denoising steps. **A.** Codon-level accuracy of generated sequences. **B.** Codon adaptation index (CAI) of generated sequences. **C.** GC content of generated sequences. **D.** Architectural hyperparameters for each model variant.

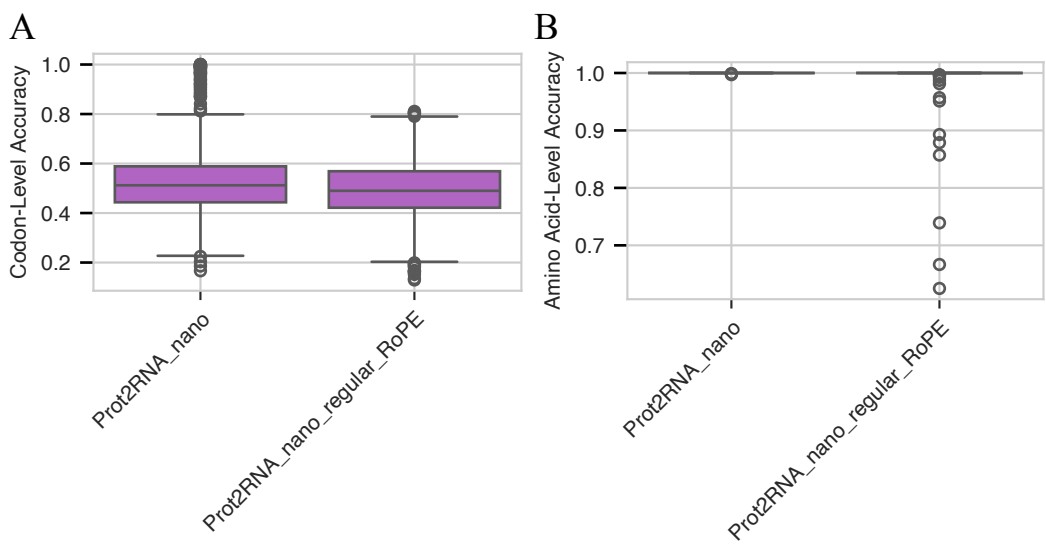

Figure 7: Comparison of standard versus modality-aligned RoPE encoding. We compare the performance of the nano-scale model using our modified RoPE (shared across protein and RNA segments) against a version with standard, independently computed RoPE (denoted as 'regular_RoPE'). Both models use a fixed cosine masking schedule and 4 denoising steps for sequence generation. **A.** Codon-level accuracy. **B.** Amino acid-level accuracy.

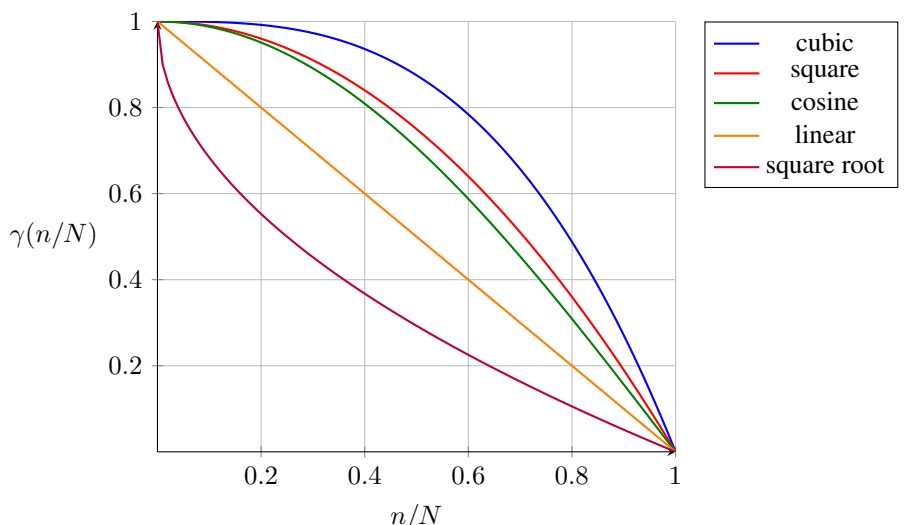

Figure 8: Mask scheduling functions $\gamma(\cdot)$ for determining masking ratio during denoising.

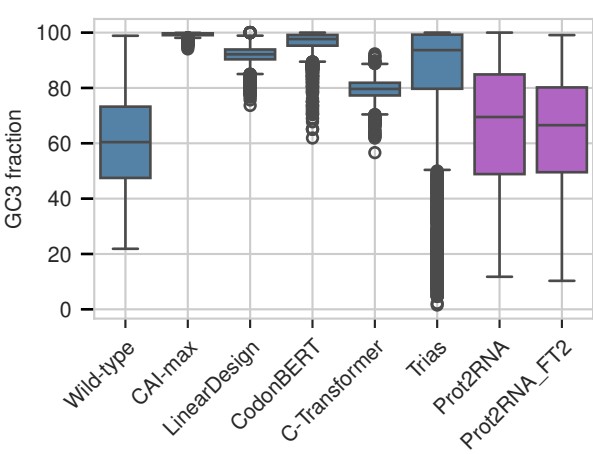

Figure 9: GC3 content of sequences generated by different methods, evaluated against wild-type references.

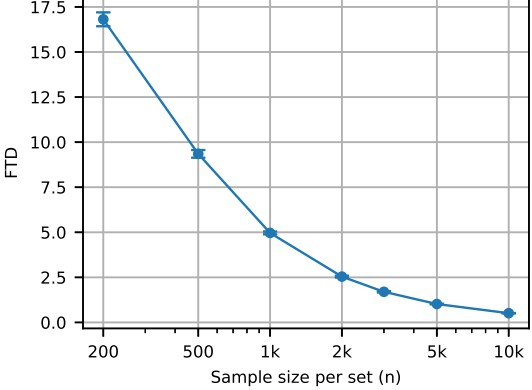

Figure 10: Sample-size dependence of the Fréchet Distance noise floor.

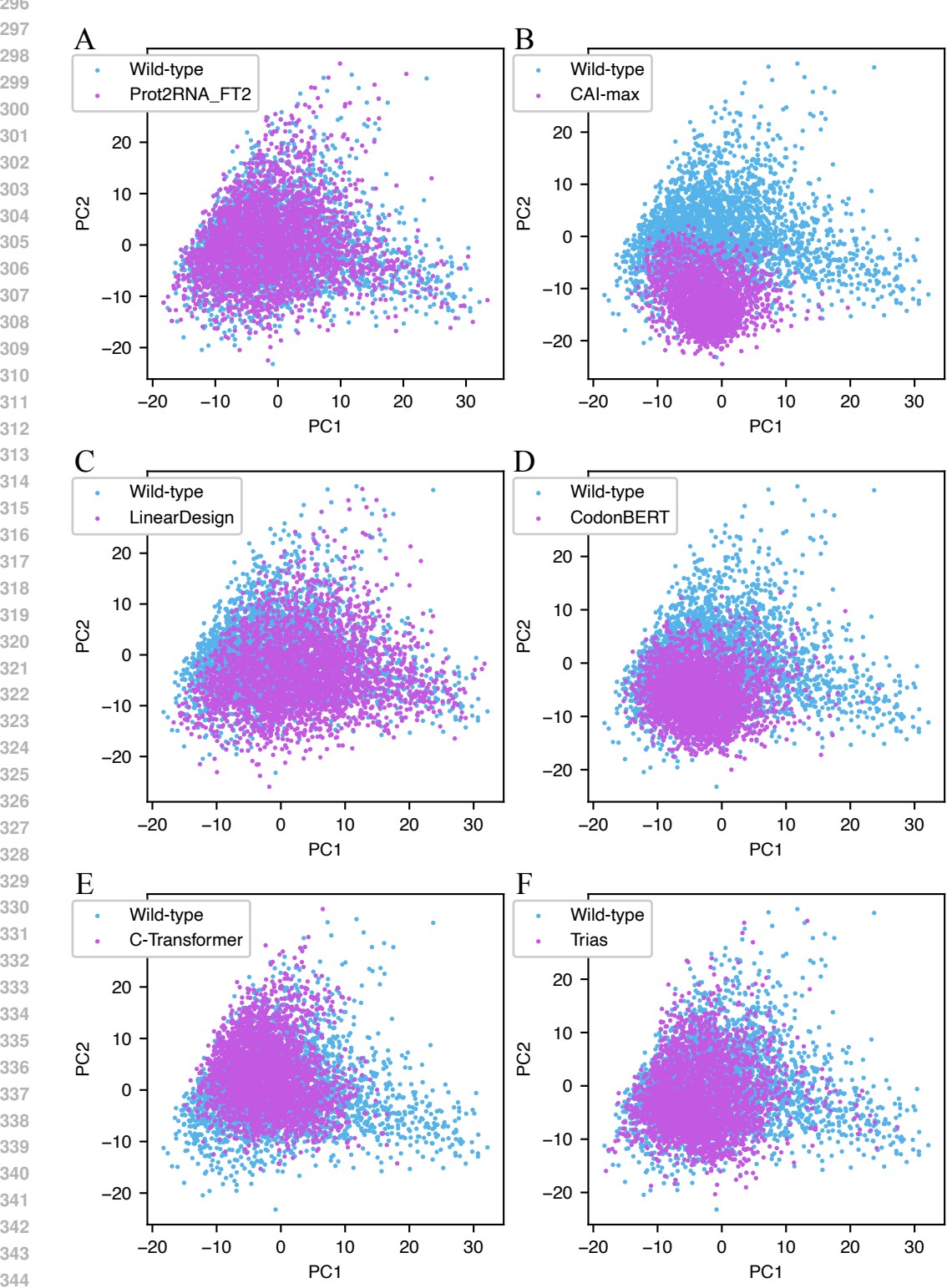

Figure 11: Principal component analysis (PCA) of sequence embeddings. Each panel shows wild-type coding sequences (light blue) compared with outputs from a given method (purple): A. Prot2RNA. B. CAI-max. C. LinearDesign. D. CodonBERT. E. CodonTransformer. F. Trias.

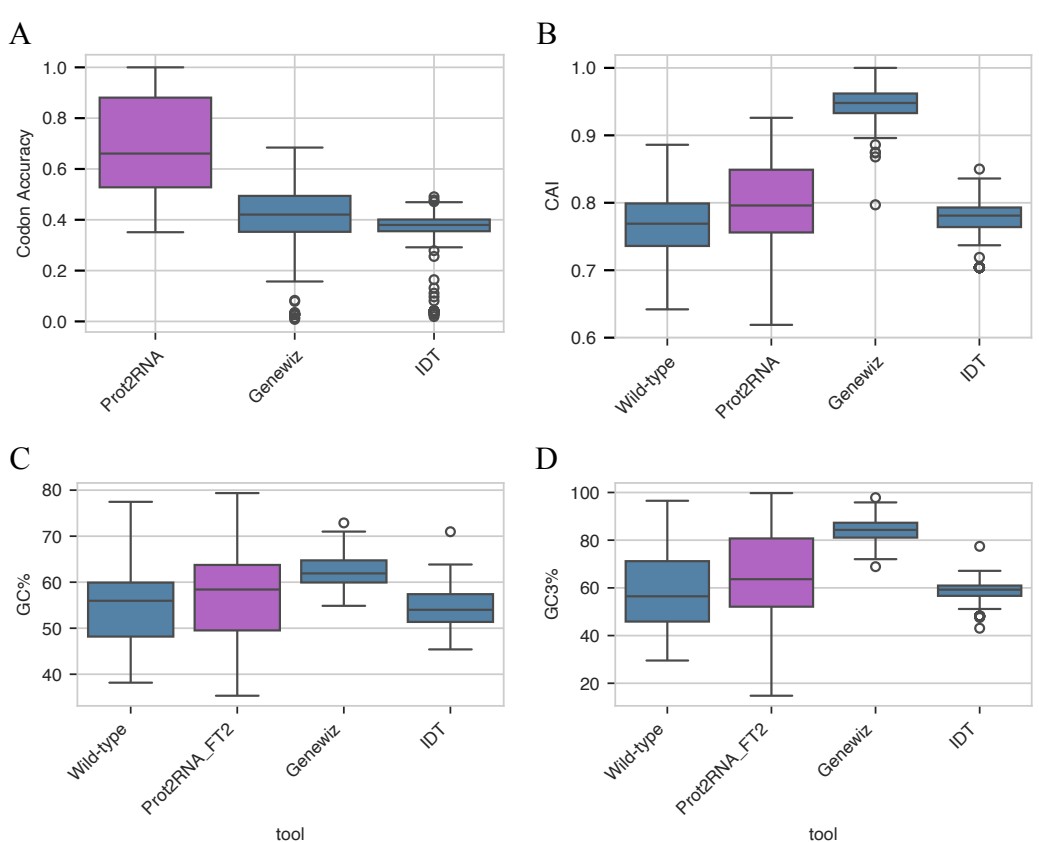

Figure 12: Comparison of Prot2RNA with commercial codon-optimization tools. A. Codon-level accuracy. B. Codon Adaptation Index (CAI). C. GC content. D. GC3 content.

