# OpenReview forum: "Prot2RNA: A Diffusion Language Model for Protein-Conditioned mRNA Coding Sequence Generation"
_ICLR.cc/2026/Conference — Submitted to ICLR 2026_

### Official Review · Reviewer_aGVJ · 2025-10-26

**Soundness:** 3
**Presentation:** 3
**Contribution:** 2
**Rating:** 2
**Confidence:** 5

**Summary:**

Here, the authors introduce Prot2RNA, a diffusion model that generates mRNA coding sequences conditioned on protein sequences. The model addresses the redundancy of the genetic code by learning how synonymous codon choices affect stability, translation, and expression, rather than relying on frequency-based optimization. Prot2RNA is trained in two stages. First, it undergoes masked diffusion pretraining on separate sets of protein and mRNA sequences to learn shared representations across modalities. Second, it is finetuned on paired protein–mRNA data, using the protein as a prompt while iteratively denoising masked RNA tokens to generate complete coding sequences. The finetuned variant (Prot2RNA-FT2) achieves the highest codon-level accuracy (0.616) among tested baselines, including CodonBERT, CodonTransformer, LinearDesign, and Trias. Generated sequences maintain GC content near 56% and CAI around 0.8, closely matching wild-type human transcripts, while avoiding over-optimization toward GC-rich codons. The authors also propose a Fréchet distance metric to measure the similarity between generated and real coding sequences. Prot2RNA yields the lowest divergence from natural mRNAs and best preserves local codon usage patterns, as shown through MinMax profile analyses.

Overall, Prot2RNA captures both global and local codon preferences relevant to expression efficiency, outperforming previous deep learning and heuristic codon optimization methods. The paper concludes that while the model does not yet integrate untranslated regions or RNA structural dynamics, it establishes a strong foundation for protein-aware mRNA design and future biologically integrated generative modeling.

**Strengths:**

-The paper provides an exceptionally clear and well-elaborated introduction, setting up the biological and computational context in a way that feels both rigorous and accessible.

-I find that the ablation studies are thorough and thoughtfully designed, giving real insight into how model choices and training strategies affect biological realism and performance.

-The authors have made the code, datasets, and model weights openly available, which is commendable and greatly helps to reproduce the work.

**Weaknesses:**

-The evaluation metrics focus primarily on codon-level accuracy and compositional similarity, which really does not directly translate to improved biological expression or experimental performance.

-The authors give limited attention to other determinants of mRNA expression, like untranslated regions, RNA secondary structure, or ribosome dynamics, which could further validate the model’s biological relevance.

-While the application is strong, the computational innovation is very incremental, as the diffusion framework largely builds on existing masked diffusion formulations without major theoretical extensions.

-The discussion of potential overfitting to synonymous codon patterns could be expanded, particularly to clarify how the model avoids simply memorizing human codon bias rather than learning transferable principles of expression.

Although Prot2RNA achieves relatively good results on computational benchmarks, it lacks experimental validation for the generated mRNA, which is my main concern. It also lacks other *in silico* biological validations including half-life, ribosomal load, etc. to better demonstrate the effectiveness of such method. The contribution would better be suited for journal with further experiments included instead of as a conference main track paper. I encourage the authors to perform experimental, wet-lab validation before submitting this work.

The authors also miss key citations:
- **Cosine mask rate scheduling:** FusOn-pLM (Vincoff, 2025);
- **Protein-conditioned mRNA design:** mRNAutils (Patel, 2025);
- **Other discrete Diffusion models:** MDLM (Sahoo, 2024)

**Questions:**

- Line 158 "high quality CDS", please add hyperlink to appendix so the definition is easier to follow

- Line 214, for pretraining the random chopping is harmless. During finetuning, if random cropping were applied inconsistently between the two modalities, could it break alignment between codons and amino acids? The authors need to justify this.

- The training setup predicts the original RNA sequence given the protein, but it’s unclear how this mechanism directly models synonymous codon substitution or optimization. Since the RNA tokens are reconstructed deterministically, the model effectively memorizes codon frequencies conditioned on amino acids rather than learning a transformation or preference over synonymous variants. If synonymous codon usage is a central goal, an analysis over synonymous sets would strengthen the claim.

- The multimodal alignment seems to rely solely on concatenation with shared positional encodings. Have the authors considered alternative fusion strategies, such as joint embedding projection or cross-attention between modalities, instead of simple concatenation?

- How easy it is to train protein vs mRNA -- do they converge as quickly?

- Line 246, the authors do not provide a clear justification for why one-step denoising suffices, nor do they compare against multi-step or continuous-time diffusion formulations that might improve sample diversity or fidelity.

- The hypothesis proposed in **Result 4.1** about codon preservation and expression level can only be validated through experiment. Naming it as "accuracy" is misleading as the metric appears to measure token-level overlap between generated and reference codons, which primarily reflects reconstruction fidelity rather than true biological quality. Furthermore, the notion of “similarity” is not rigorously defined. It remains unclear whether the model produces novel codon variants or merely regenerates sequences from the training distribution. An analysis of sequence novelty would be required to substantiate this claim.

- The statement in **Result 4.2** *"reflecting natural codon usage where high expression is not driven solely by codon bias, but a range of other factors, as previously shown"* contradict the hypothesis proposed in **Result 4.1** that capturing codon preservation does predict expression.

- For **Result 4.3** the proposed benchmark metrics is problemetic as the [CLS] tokens from different models reside in distinct latent spaces with incompatible scales and semantics. The authors should either project all sequences into a shared encoder space or clarify how inter-model embedding distances can be meaningfully interpreted.

- For **Result 4.4**, $\mathrm{MINMAX}$ measures the difference in codon frequency distributions relative to highly expressed transcripts, not true codon optimization or functional adaptation. Lower distances could reflect memorization of training statistics rather than meaningful synonymous substitutions, raising concerns that the metric conflates similarity with optimization. Clarifying whether this measure capture genuine adaptation or mere distribution alignment and overfit would strengthen the claims.

---

> ### Author Response · Authors · 2025-11-21
>
> We are grateful to the reviewer for the constructive and detailed feedback, as well as the positive comments on the clarity, ablations, and reproducibility of the study. Our responses to the specific points are provided below.
>
> **W1.** We agree that codon-level accuracy and compositional metrics do not directly reflect biological expression, and in the future revision we will frame them strictly as sequence-fidelity measures quantifying how human-like the generated CDSs are, rather than as functional predictors. To explore whether expression-related evaluation could be performed computationally, we tested several sequence-only models (fine-tuned CodonBERT [1,2], AIDO.RNA [3], RiboNN [4]), even though they publish correlation ranges of approximately 0.28-0.67. When we applied these models to the ground-truth sequences in our test set, their predictions did not exhibit a consistent separation between the two groups, likely due to the combination of moderate predictive correlations and the fact that these models quantify distinct biological endpoints (for example, TE prediction depends heavily on 5′UTR context, which RiboNN uses as part of its input). As they do not offer a unified or task-aligned evaluation criterion in this setting, we avoid interpreting these outputs as validation and instead highlight experimental measurement as the appropriate next step.
> We would welcome any suggestions for computational proxies the reviewer considers meaningful prior to wet-lab evaluation.
>
> **W2.** We fully agree that untranslated regions, RNA secondary structure, and ribosome dynamics play critical roles in determining mRNA stability and translational efficiency. Prot2RNA, however, is designed specifically to model coding-region synonymous variation, and therefore operates only on CDS sequences in this work. Incorporating UTRs or structural dynamics would require a substantially different modeling setup, including different tokenization, conditioning mechanisms, and training data, and is outside the scope of the present study.
> We made this limitation explicit in the revised manuscript: the Conclusion section highlights that Prot2RNA does not currently model UTRs, folding behavior, or ribosomal kinetics, and that integrating these determinants is an important direction for future work. We agree that such extensions could significantly enhance the biological relevance of the approach and plan to explore them in subsequent iterations of the model.
>
> **W3.** We agree that our diffusion formulation follows the same discrete-time masked denoising framework introduced in LLaDA [5], and we do not claim theoretical novelty in the diffusion mechanism itself. Our contributions are instead application-driven, which aligns with the paper’s declared primary area (“applications to physical sciences”).
> Specifically, we adapt masked diffusion to a multimodal biological setting where amino acids and codons must be aligned positionally, and where the corruption process selectively targets only the RNA modality during finetuning. This design enables the model to learn fine-grained, context-dependent synonymous codon usage patterns and to generate biologically plausible mRNA sequences conditioned on unseen protein sequences, capabilities that prior diffusion LMs were not developed or evaluated for.
>
> **W4.** In the current study, we generate multiple sequence variants per protein using different decoding configurations (masking schedules and denoising step counts), which produce distinct synonymous realizations while preserving overall naturalness. These variants consistently maintain appropriate GC/GC3 composition and codon-usage profiles, indicating that the model is sampling within the space of plausible human-like codon choices rather than reconstructing a single template. A more explicit analysis of novelty within synonymous sets and across multiple generated samples (by applying temperature, top-k, or nucleus sampling) is a valuable next step, and we plan to explore this in future work.
>
> **W5.** We agree that experimental measurements, such as mRNA half-life, ribosomal occupancy, or direct protein-expression assays, would significantly strengthen the biological validation of our approach. While these experiments are beyond the scope of the present submission, we plan to pursue them in subsequent work, and we now explicitly acknowledge this as a major limitation in the revised manuscript. As clarified in the Conclusion, wet-lab validation represents the most important next step for assessing the functional properties of Prot2RNA-generated sequences. We appreciate the reviewer’s perspective and agree that a more comprehensive biological evaluation will be needed in future extensions of this research.

---

> ### Author Response · Authors · 2025-11-21
>
> **W6.** We have added a dedicated paragraph in the Related Work section summarizing iterative refinement and discrete diffusion models (including MDLM).
> We were not able to find the reference “mRNAutils (Patel, 2025)”  and would be grateful if the reviewer could provide the full title or link so we may review and include it appropriately.
> Regarding FusOn-pLM, while we acknowledge it as a protein language model that employs cosine-scheduled masking, its focus on fusion oncoprotein representation learning appears orthogonal to our setting of protein-conditioned mRNA generation and discrete diffusion. If the reviewer believes there is a specific conceptual connection we should emphasize, we would appreciate further clarification.
>
> **Q1.** We clarified the definition of “high-quality CDS” directly in the main text and made it explicit that Appendix Section A provides full details for the entire preprocessing procedure.
>
> **Q2.** Random cropping is applied only during pretraining on single-modality sequences to fit within the context window and does not affect paired alignment. During finetuning, we explicitly restrict training to protein-CDS pairs shorter than 1,022 amino acids and codons, ensuring that both sequences fit entirely within the 2,048-token window without any cropping. This is stated in the manuscript (Section “Finetuning Stage”).
>
> **Q3.** Our training objective reconstructs natural coding sequences given their corresponding protein sequences, which teaches the model the statistical structure of human synonymous codon usage in a multimodal setting. However, we do not claim that this objective explicitly optimizes or ranks synonymous variants. As clarified elsewhere, Prot2RNA aims to generate human-like CDSs, and the reconstruction task serves as a way to learn natural codon distributions conditioned on protein context, rather than a deterministic mapping or an expression-optimized transformation.
> We agree that this objective primarily captures conditional codon frequency patterns, not functional optimization, and that analyses over synonymous codon sets could provide additional insight into how the model distributes probability mass within each synonymous group. This is an interesting direction for future work, particularly as part of a larger effort to evaluate synonymous novelty and deviation from the training distribution once reliable functional benchmarks are available.
>
> **Q4.** In this work, we adopt concatenation with shared positional encodings because it provides a simple and effective way to align amino acids with their corresponding codons and allows the model to jointly process both modalities without introducing additional architectural complexity. This choice is consistent with prior multimodal language-modeling work where the modalities share a common token space or where one modality is conditionally predictive of the other.
> We agree that more elaborate fusion mechanisms, such as cross-attention, joint projection layers, or modality-specific conditioning blocks, are promising alternatives that could potentially capture richer interactions between protein and RNA representations. Exploring these options would require additional architectural design and hyperparameter tuning and is outside the scope of the present study, which focuses primarily on demonstrating the feasibility and biological utility of protein-conditioned CDS generation.
>
> **Q5.** We thank the reviewer for the question. In the pretraining stage, protein and mRNA sequences are trained under the same masked-diffusion objective, but their loss curves differ slightly due to inherent differences in vocabulary structure and sequence statistics. Proteins have a smaller and more evenly distributed 21-token alphabet, whereas RNA uses a 64-codon vocabulary with more skewed usage frequencies. As a result, the protein loss typically decreases marginally faster and to a slightly lower value than the RNA loss.
> Importantly, both modalities converge stably within the pretraining schedule, and we did not observe optimization difficulties that impacted finetuning or downstream generation. Because these differences were small and did not affect the main conclusions, we did not include the training curves in the manuscript, but we are happy to provide them upon request.
>
> **Q6.** We thank the reviewer for this question. Our one-step denoising objective follows the standard training paradigm for discrete diffusion language models, as adopted in LLaDA (Nie et al., 2025). In this framework, the model learns to predict the fully denoised sequence $x_0$  from a partially masked input $x_t$  sampled at a random corruption level ${t \sim \mathcal{U}(0,1)}$. This single-step objective is equivalent to training across all possible noising levels in expectation and has been shown to yield stable and efficient learning for discrete sequences.

---

> ### Author Response · Authors · 2025-11-21
>
> **Q7.** Codon-level accuracy in our work refers strictly to token-level overlap with the natural CDS for the same protein and is used only as a reconstruction fidelity metric, not as an indicator of biological expression. We acknowledge that “accuracy” in this sense does not imply functional optimization, and we will ensure this is clearly distinguished from biological performance in future revisions.
> We agree that codon accuracy does not capture whether the model produces novel synonymous variants or simply aligns closely with natural patterns. While our current evaluation includes complementary distribution-level metrics (MinMax, GC/GC3, embedding-based Fréchet distance), a more explicit analysis of synonymous novelty would indeed strengthen the characterization of the model’s generative behavior. We view this as a valuable direction for future work.
>
> **Q8.** We agree that codon-level accuracy and codon preservation should not be interpreted as evidence of high expression, and we do not intend to make that claim, and we will make sure to correct the narrative in the future revisions of the manuscript. Result 4.1 evaluates sequence fidelity (how closely the generated CDS resembles natural human coding patterns), not functional expression. Result 4.2 reiterates a well-established biological result: expression levels depend on many factors beyond codon bias.
>
> **Q9.** All embeddings used for computing the Fréchet distance and the PCA visualization are extracted from the same pretrained encoder-only CodonBERT model (Li et al., 2024), applied consistently to both wild-type and generated sequences. Thus, the embeddings reside in a shared latent space with consistent scale and semantics, and inter-model distances are computed within this common representation. We do not compare [CLS] embeddings originating from different model architectures.
> To prevent confusion, we have updated the manuscript to explicitly state that the CodonBERT encoder used for evaluation is a separate, fixed model independent of the CodonBERT generative baseline included in our comparisons.
>
> **Q10.** We agree that MinMax distance measures distributional similarity to highly expressed transcripts and should not be interpreted as evidence of functional optimization. In our work, MinMax is used only as a sequence-fidelity metric, that is, to assess whether the generated coding sequences follow human-like positional codon usage patterns, not as a proxy for expression or functional adaptation.
> We have added Appendix Section D (“Evaluation Metrics”) to the revised manuscript which explains all the used metrics in more detail and we hope this clarifies MinMax as well.
>
> ---
>
> [1] Li, S., Moayedpour, S., Li, R., Bailey, M., Riahi, S., Kogler-Anele, L., ... & Jager, S. (2024). CodonBERT large language model for mRNA vaccines. Genome research, 34(7), 1027-1035.
>
> [2] Li, S., Noroozizadeh, S., Moayedpour, S., Kogler-Anele, L., Xue, Z., Zheng, D., ... & Jager, S. (2025). mRNA-LM: full-length integrated SLM for mRNA analysis. Nucleic Acids Research, 53(3), gkaf044.
>
> [3] Zou, S., Tao, T., Mahbub, S., Ellington, C. N., Algayres, R., Li, D., ... & Xing, E. P. (2024). A large-scale foundation model for rna function and structure prediction. bioRxiv, 2024-11.
>
> [4] Zheng, D., Persyn, L., Wang, J., Liu, Y., Ulloa-Montoya, F., Cenik, C., & Agarwal, V. (2025). Predicting the translation efficiency of messenger RNA in mammalian cells. Nature biotechnology, 1-14.
>
> [5] Nie, S., Zhu, F., You, Z., Zhang, X., Ou, J., Hu, J., ... & Li, C. (2025). Large language diffusion models. arXiv preprint arXiv:2502.09992.

---

> > ### Comment · Reviewer_aGVJ · 2025-11-21
> >
> > Thank you to the authors for putting together a thoughtful and careful response! I appreciate the clarifications on masking, positional encoding, and framing codon accuracy purely as a fidelity metric. The manuscript reads more clearly after these adjustments, and the added context around limitations is helpful.
> >
> > That said, my main hesitation still remains. Even with the expanded discussion, it is hard for me to judge the biological relevance of the model without some functional validation. Right now every metric is still a sequence-level or distribution-level surrogate, so I do not yet have evidence that Prot2RNA learns principles that matter for expression, stability, or translation. I do not expect full experiments for a conference submission, but even lightweight *in silico* functional checks would go a long way. For example, half-life, ribosome load, or translation-rate predictors of the type used in mRNAutilus (here is the paper that should be cited: https://openreview.net/forum?id=2gLAV6ROxQ) could give me at a bit of a directional sense of whether the generated sequences look more or less functional. These are imperfect models, but they test the core hypothesis directly, which is what I am missing most.
> >
> > I also still have questions about what the model is actually doing within each synonymous codon set. Since the fine-tning objective is reconstruction-based, it is difficult for me to know whether the model is learning meaningful patterns or maybe just memorizing human codon distributions. A simple analysis of within-synonym diversity, novelty relative to the training set, or how often the model prefers rare versus common codons would help me understand how Prot2RNA behaves. At the moment, I cannot tell if it is capable of producing sequences that differ from the native pattern in any controlled or interpretable way.
> >
> > Also, the classifiers used in mRNAutilus for half-life, ribosomal load, and translation rate would be good candidates for proxies. You do not need to compare to that work directly. Even a brief appendix analysis using any of these predictors would strengthen the paper by tying the generated sequences to something biological rather than only distributional. This would make it much easier for me to assess the value of protein-conditioned mRNA generation in practice.
> >
> > I appreciate the detailed rebuttal, and I hope these suggestions are helpful as you continue to refine the work. If these remaining points are clarified, I will raise my score!

---

> > > ### Comment · Reviewer_aGVJ · 2025-11-26
> > > **Happy to clarify.**
> > >
> > > I'd like to continue the conversation with the authors and I am happy to raise my score if they can provide these final clarifications. :)

---

### Official Review · Reviewer_SWB8 · 2025-10-30

**Soundness:** 2
**Presentation:** 2
**Contribution:** 2
**Rating:** 2
**Confidence:** 5

**Summary:**

This paper introduces Prot2RNA, a diffusion-based transformer model that given a protein sequence, it generates mRNA coding sequences. The model is trained in two stages- firstly, a masked diffusion pre-training on protein and mRNA sequences separately followed by fine-tuning on paired protein-mRNA data with high expressing pairs. In-silico experiments are conducted to compare Prot2RNA with other baselines to evaluate its benefits. The authors claim and demonstrate that Prot2RNA generates biologically realistic coding sequences that align well with respect to metrics such as codon adaptation index (CAI) and GC content.

**Strengths:**

1. The codon optimization problem being studied in the paper is relevant for the bioinformatics community and for mRNA based drug design. The combinatorial space of mRNAs coding for same protein makes it costly to find the best mRNA experimentally. Utilising data-driven ML methods for this problem is quite sensible.
2. The proposed method technically is sensible and given the general pre-training, the fine-tuning stage make sense. The chosen baselines are also adequate and the presented results with respect to CAI and GC content are convincing.
3. The paper is well-written and easy to follow.

**Weaknesses:**

Although the paper describes the motivation, methodology and results clearly, there are some major gaps which I describe below.

1. **Weak Biological/Mechanistic Plausibility and Unsubstantiated Claim:**

a) The paper’s core claim, positioned in the introduction, is that it overcomes the limitations of traditional codon optimization methods (such as optimizing Codon Adaptation Index) by capturing codon-expression relationships that these metrics fail to fully encapsulate. While the paper reports improved metrics such as CAI and GC content, demonstrating in-silico biological plausibility, it critically fails to provide any novel, biologically-grounded evidence that its method has actually learned the complex, non-linear codon-expression relationships it claims to model. The same argument the authors use against traditional methods—that optimizing CAI does not guarantee high expression—holds true for this work. From a biology standpoint as well, it is a well-known fact that CAI and other metrics are known to be weakly correlated with translational efficiency (see [1], [2] as some examples for well-established evidence).

b)  Related to (a), the argument that fine-tuning on high-expression mRNA-protein pairs will enforce the model to learn to predict high-expressing mRNAs is not guaranteed. In general, during fine-tuning, the degree to which a large pre-trained model adapts to a new, specific task (like predicting high expression) is dependent on factors such as the size and distribution of the fine-tuning dataset, the magnitude of the learning rate, and the degree of catastrophic forgetting of the original pre-training task (see [3], [4], [5], [6], [7] for few works demonstrating this). The model's adaptation is an empirical outcome, not a theoretical certainty. Therefore, without a strong theoretical or empirical explanation that goes beyond simply fine-tuning, the claim remains an unsubstantiated hypothesis. I understand that empirical validation is near impossible without doing any wet-lab experiments but that is a major problem with multiple prior works (including some of the baselines mentioned in this work). Few works such as LinearDesign have used wet-lab experiments to substantiate their claim and without it this paper just becomes another paper which claims to produce high expression mRNA without any evidence for the same. I believe adding this would substantially improve the paper for next version. In the absence of this, providing strong theoretical guarantees for the impact of fine-tuning would be helpful. The authors have not shown what is learned, only that the generated sequences score well on the same low-correlation metrics they themselves rightly criticize.

2.  **Limited Methodological Novelty and Unclear Performance Gains:**

a) The paper references a “masked diffusion” framework but lacks a clear probabilistic formulation (e.g., noise schedule, Markov process definition unless I missed it) but it appears to me that the proposed masked diffusion process appears to be a direct and straightforward adaptation of works such as LLaDA (Nie et al., 2025), DPLM (Wang et al., ICML 2024) applied to discrete sequences. The novelty is limited to applying this known machine learning paradigm directly to the task of codon optimization.

b) Moreover from an algorithmic standpoint, the use of a diffusion model, which predicts masked tokens iteratively (a key difference from single-shot prediction models like traditional Masked Language Models such as CodonTransformer or Autoregressive Transformers), does not automatically translate to performance gains in sequence generation. Looking broadly at the NLP literature, currently the benefits of diffusion transformers over standard Autoregressive (AR) or Masked Language Model (MLM) based transformers in general Language Modeling are an area of active research, and diffusion models have not shown substantial performance gains over MLM or AR transformers for discrete data, unlike for continuous data (Although slightly old, but still relevant blogpost [7] and other works such as [8], [9], [10]). So theoretically, I wonder if Prot2RNA performing better compared to non-diffusion transform counterparts such as CodonTransformer is because of difference in model architecture, pre-training and fine-tuning data rather than because of mechanics of diffusion itself. The authors fail to provide a conceptual or theoretical reason why the iterative refinement of a diffusion model is particularly well-suited for the biophysical constraints of codon optimization, as opposed to a single-shot prediction. In works such as LLaDA as well which the authors build on, the AR and diffusion models are evaluated under same protocol to make the comparison fair and there as well, it is not always empirically clear that diffusion transformer models are always better than non-diffusive transformer models and often performs comparably.

c) Furthermore, fine-tuning stage for codon optimization is also proposed in CodonTransformer where they also fine-tune the model on highly expressing genes. If the architecture type (e.g., Transformer), pre-training data and scale, and fine-tuning data were equalized between this diffusion-based approach and a non-diffusion model like CodonTransformer, I wonder if the iterative diffusion process would offer any significant performance benefit. This could act as an ablation to motivate the use of diffusion models empirically since the only difference from prior works such as CodonTransformer is using a diffusion model for multi-step denoising over conceptually single-step denoising with MLM/AR based transformers.


3. **Issue with experiment on Codon-Level Accuracy:** This sequence-level experiment/metric is somewhat trivial and can be misleading. Codon-level accuracy (matching wild-type codons) is an inadequate metric because the genetic code is synonymous. Exact codon matching is unnecessary for function and does not correlate with expression. This metric can primarily reward memorization or overfitting to the training data and can unfairly penalize models such as LinearDesign which prioritize functional (structural) features over sequence identity, thus confounding model quality with design intent.

The authors define the test set as "highly expressed native sequences" and assert that high codon overlap with these sequences implies high expression potential. I fear that this is a circular and self-fulfilling prophecy: the model is rewarded for statistically reproducing high-expression patterns already present in the data, not for demonstrating an independent, generalizable ability to predict expression. Without a control experiment that correlates codon accuracy with an independent measure of expression (e.g., protein yield), this metric is inconclusive and favors memorization over generalization.

[1] Link Between Individual Codon Frequencies and Protein Expression: Going Beyond Codon Adaptation Index, 2024

[2] Limitations of codon adaptation index and other coding DNA-based features for prediction of protein expression in Saccharomyces cerevisiae, 2004

[3] On the Importance of Data Size in Probing Fine-tuned Models, ACL 2022

[4] Unveiling the Secret Recipe: A Guide For Supervised Fine-Tuning Small LLMs, 2024

[5] An Empirical Study of Catastrophic Forgetting in Large Language Models During Continual Fine-tuning, 2023

[6] Revisiting Catastrophic Forgetting in Large Language Model Tuning, EMNLP, 2024

[7] Diffusion language models, https://sander.ai/2023/01/09/diffusion-language.html

[8] Large Language Diffusion Models, ICML 2025

[9] DiffuSeq: Sequence to Sequence Text Generation with Diffusion Models, ICLR 2023

[10] Latent Diffusion for Language Generation, NeurIPS 2023

**Questions:**

The following questions are a reformulation of what I described in Weaknesses section.

1. Fine-tuning on highly expressed mRNA-protein pairs is claimed to improve generation of high-expression sequences. Could you explain theoretically or empirically how fine-tuning guarantees this adaptation?
2. Given that CAI and GC content are known to correlate weakly with expression, how do you justify using these metrics as primary evidence for biological relevance? Could the observed improvements be merely reproducing codon usage frequencies rather than capturing true expression determinants?
3. The masked diffusion framework appears similar to prior work in LLaDA or DPLM. Can you clarify what is novel about your diffusion formulation specifically for codon optimization?
4. Iterative diffusion over masked tokens is proposed as a key architectural choice. Could you provide a conceptual or theoretical justification for why this multi-step refinement is particularly suited to modeling codon constraints or translation efficiency, as opposed to single-shot MLM or autoregressive models like CodonTransformer given that in language modeling diffusion transformers are known to perform comparably to non-diffusion transformers?
5. Given that the key difference between Prot2RNA and baselines such as CodonTransformer is diffusion transformer vs non-diffusion transformer and since prior works in NLP have shown under identical settings, these models perform comparably, did you perform any controlled ablation comparing Prot2RNA to a standard Transformer (e.g., CodonTransformer) using identical pre-training, architecture size, and fine-tuning data? If so, what performance gains are directly attributable to the diffusion process rather than other confounding factors?
6. Codon-level accuracy is used as a metric, but exact codon matching is not biologically necessary and may favor memorization over functional generalization. Can you provide any results showing that higher codon-level accuracy actually correlates with translation efficiency or protein expression?
8. The test set consists of highly expressed native sequences. How do you ensure that high codon overlap with this set is not simply rewarding memorization or reproducing training distribution biases? Have you considered testing on out-of-distribution proteins or low-expression sequences to validate generalization?
9. For models like LinearDesign that optimize structural or stability features rather than codon identity, codon-level accuracy penalizes their objective. How do you justify the metric comparison across models with fundamentally different design goals? Would functional metrics (e.g., translation efficiency, mRNA stability) provide a more meaningful comparison?
10. Have the authors considered wet-lab validation or in-vitro translation assays to substantiate claims regarding high-expression mRNA generation? Even small-scale experiments could strengthen the biological claim substantially and would differentiate the work as methods-wise, this work is rather similar to prior works as discussed in Weaknesses section.
11. How sensitive are Prot2RNA’s results to the choice of fine-tuning dataset size or masking schedule? Could differences in these hyperparameters explain the reported performance improvements rather than the diffusion modeling per se?

---

> ### Author Response · Authors · 2025-11-21
>
> We appreciate the reviewer’s careful reading of our work and the positive remarks on its relevance, technical approach, and clarity. As noted, the questions restate the issues raised in the Weaknesses section; we address each of these points in order below.
>
> **Q1.** Our second fine-tuning stage is intended to bias the model toward producing CDS patterns more similar to those of highly expressed sequences, but we agree that we  cannot claim that it guarantees improved expression. At present, our evaluation focuses on sequence fidelity rather than functional optimization.
>
> **Q2.** We agree that CAI and GC content correlate only weakly with expression and therefore do not use them as direct measures of translational efficiency. Instead, we report them as compositional sanity checks to ensure that generated sequences remain within biologically realistic ranges observed for highly expressed human mRNAs. In particular, GC content provides an indirect proxy for mRNA stability, while excessively low CAI values can indicate suboptimal codon usage that may slow translation. Our results show that Prot2RNA reproduces these properties at moderate, biologically consistent levels rather than maximizing them, suggesting that the model captures broader codon-usage regularities beyond simple frequency matching.
>
> **Q3.** We agree that the masked diffusion process follows the same discrete-time corruption and denoising formulation as LLaDA [1]. Our novelty is application-specific: we extend this paradigm to a multimodal biological domain where each amino acid token aligns with a codon triplet, requiring modality-aligned positional encodings and a corruption process that affects only one modality. We further show that this adaptation enables realistic protein-conditioned mRNA generation capturing codon-level biological signals, something prior diffusion LMs were not designed or tested to do.
>
> **Q4.** Our use of iterative masked diffusion is motivated by the structure of the codon-generation task. Codon selection is a high-degeneracy mapping where many synonymous codons are valid, and global sequence properties (GC balance, positional codon patterns, local compositional constraints) depend on the entire CDS rather than on a single position. Single-shot MLMs make one pass and cannot revise globally inconsistent choices, and autoregressive decoders impose an arbitrary left-to-right order on a process that is biologically non-directional.
> In contrast, the diffusion formulation allows the model to iteratively revise uncertain positions based on full bidirectional context at every step. In our ablations, a small number of refinement steps improved sequence-level coherence and stabilised GC/GC3 and MinMax profiles (Appendix Table 2), after which gains saturated. This makes iterative refinement a practical fit for a task where many positions must be adjusted jointly rather than decided once.
> We have added a brief explanation of this rationale to Section 3.1.
>
> **Q5.** We agree that a controlled architectural ablation comparing diffusion and non-diffusion Transformers under identical pretraining and finetuning conditions would more directly isolate the effect of the diffusion mechanism. In the current work, our focus is primarily on the application side - demonstrating that a diffusion-based model can perform protein-conditioned CDS generation, rather than on establishing the intrinsic superiority of diffusion over standard Transformers. Accordingly, we do not claim that the improvements we observe arise solely from the diffusion process.
> We note that constructing a fully controlled comparison would require re-implementing a non-diffusion model with the same multimodal alignment strategies, corruption scheme, training objective, tokenizer, and data pipeline used in Prot2RNA. Such a redesign would entail substantial architectural changes and is closer to a separate project than a small ablation.
> We would welcome the reviewer’s suggestions on which components they consider most important for a controlled comparison, and we view broader architectural studies as a valuable direction for future work.

---

> ### Author Response · Authors · 2025-11-21
>
> **Q6.** We fully agree with the reviewer that codon-level accuracy does not necessarily correlate with expression. In future revisions of the manuscript, we will clarify that we use this metric only to assess naturalness, not functional efficiency.
> To explore whether expression-related signals could be assessed computationally, we tested several sequence-only predictors (e.g., fine-tuned CodonBERT [2, 3], AIDO.RNA [4], RiboNN [5]). These predictors target different biological quantities (transcript abundance, protein abundance, or translation efficiency) and their published correlations with experimental measurements are moderate (roughly 0.28–0.67 across tasks). Applied to the ground-truth sequences in our dataset, these models did not yield a clear high-/low-group separation, which is consistent with their differing biological objectives and the influence of non-CDS features on some endpoints (e.g., 5′UTR contributions to TE). Since they do not provide a coherent evaluation signal for this task, we do not rely on them here and instead emphasize the need for direct experimental validation.
> We welcome any suggestions for computational strategies the reviewer finds appropriate.
>
> **Q7.** We thank the reviewer for this question. Our test set is designed to evaluate generalization to novel proteins by clustering all proteins at 80% sequence identity and assigning entire clusters to different splits. This ensures that no test protein appears in the training data and prevents direct memorization of protein-CDS pairs.
>
> **Q8.** We agree that codon-level accuracy is not an appropriate measure of functional quality for models. In our evaluation, codon accuracy is used only to quantify sequence fidelity, how closely a generated CDS resembles natural human coding sequences, not to assess the internal goals of structure-focused models, and we will clarify this in future revisions of the manuscript.
>
> **Q9.** We thank the reviewer for raising this important point. We fully agree that wet-lab assays, including in-vitro translation and mRNA stability measurements, would provide the strongest evidence for the biological relevance of Prot2RNA-generated sequences. While such experiments are planned as part of our future work, they were not feasible within the current scope. We have now made this limitation explicit in the revised manuscript: the Conclusion section clearly states that functional validation is the next key step for establishing the expression impact of Prot2RNA-designed transcripts. We appreciate the reviewer’s suggestion and intend to pursue these experiments in follow-up work.
>
> **Q10.** We thank the reviewer for the insightful comment. We did not have time, but will definitely check whether diffusion vs. MLM pretraining with a single-step denoising yields any boost in performance.
>
> ---
>
> [1] Nie, S., Zhu, F., You, Z., Zhang, X., Ou, J., Hu, J., ... & Li, C. (2025). Large language diffusion models. arXiv preprint arXiv:2502.09992.
>
> [2] Li, S., Moayedpour, S., Li, R., Bailey, M., Riahi, S., Kogler-Anele, L., ... & Jager, S. (2024). CodonBERT large language model for mRNA vaccines. Genome research, 34(7), 1027-1035.
>
> [3] Li, S., Noroozizadeh, S., Moayedpour, S., Kogler-Anele, L., Xue, Z., Zheng, D., ... & Jager, S. (2025). mRNA-LM: full-length integrated SLM for mRNA analysis. Nucleic Acids Research, 53(3), gkaf044.
>
> [4] Zou, S., Tao, T., Mahbub, S., Ellington, C. N., Algayres, R., Li, D., ... & Xing, E. P. (2024). A large-scale foundation model for rna function and structure prediction. bioRxiv, 2024-11.
>
> [5] Zheng, D., Persyn, L., Wang, J., Liu, Y., Ulloa-Montoya, F., Cenik, C., & Agarwal, V. (2025). Predicting the translation efficiency of messenger RNA in mammalian cells. Nature biotechnology, 1-14.

---

### Official Review · Reviewer_jC16 · 2025-11-01

**Soundness:** 2
**Presentation:** 2
**Contribution:** 2
**Rating:** 4
**Confidence:** 3

**Summary:**

This paper developed diffusion language models (DLM) for human mRNA coding sequence generation. The generated sequences preserve similar codon patterns as in highly expressed natural transcripts for target proteins. Following the architecture design of LLaDa, a well-known DLM, the authors conducted two-stage training after curating a high-quality human protein-mRNA pair dataset. Evaluations by some biologically meaningful metrics proved the efficacy of trained DLM in approximating the distributions of highly expressed codons.

**Strengths:**

1. The authors constructed a high-quality dataset comprising over 100,000 human protein–mRNA sequence pairs. The models were trained exclusively on human data without incorporating signals from other species.
2. The study adapted the architecture of state-of-the-art discrete diffusion language models for mRNA codon generation.
3. The paper is well organized, and the flow of ideas is generally coherent. There are not many grammatical errors throughout the manuscript.

**Weaknesses:**

1. Some potential baselines are missing.

- It is not clearly stated in the paper why the paper adopted parallel diffusion models rather than autoregressive (AR) GPT-like paradigm. Could you please explain the reasons and possibly add a AR baseline comparison if plausible?

- It seems that the baselines listed in the paper are not deep generative models which approximate the distribution of mRNA codon. E.g, CodonBERT is an understanding model using cross-attention mechanism to align protein and codons, which is similar to CLIP. Although I agree that most existing work about applying diffusion models or autoregressive models for biological sequence generation is about non-coding RNA (ncRNA)/ UTR or even DNA [1][2], it would be good to explore whether these methods can be also used for mRNA generation with certain changes on tokenizers (possibly nucleotide -> codons) like the authors said in line 100-102.

2. The goal of coding sequence generation is not clear to me. The paper says "synthesize new coding sequences that maximize expression efficiency". But I am confused by the metrics used in the paper. It seems that the metrics like CAI or codon-level accuracy mostly reflect how close the generated sequences and wild natural highly expressed sequences are in terms of codon distributions. Is the goal to be closer to natural sequence distribution or maximize the expression efficiency?

   If it is latter, would it be possible to provide some direct proxies of expression efficiency as metrics like Mean Ribosome Loading (MRL) [4] and Translation Efficiency (TE) [5] measured on UTR sequences?

3. Some statements about experiment results are not sound or well supported by the evidence.
- In line 306-307, it is said that “This improvement underscores the benefit of finetuning on highly expressed sequences.” But the retrieval improvement from 0.612 to 0.616 does not convince me.
- Regarding the CAI results, would the model be better if it has higher CAI values or resembles CAI distribution of wild sequences?
- The authors gave detailed analyses to every method including different baselines. But I feel like the strengths of the proposed new method should be highlighted more and comparisons with other methods could be condensed. For now, the analyses about CAI and GC are not that informative.
- In the conclusion section, the authors mentioned "biologically relevant balance between expression optimization and sequence fi-
delity." It is not clear which metrics are reflective of expression optimization and which metrics are indicative of sequence fidelity.


[1] Zhao, et al. GenerRNA: A generative pretrained language model for de novo RNA design.

[2] Zhang, et al. RNAGenesis: A Generalist Foundation Model for Functional RNA Therapeutics.

[3] Li, et al. Latent Diffusion Model for DNA Sequence Generation.

[4] Sample, et al. Human 5’ utr design and variant effect prediction from a massively parallel
translation assay. Nature biotechnology, 37(7):803–809, 2019.

[5] Cao, et al. High-throughput 5’ utr engineering for enhanced protein production in non-viral gene therapies. Nature
communications, 12(1):4138, 2021.

**Questions:**

1. Due to limit of context window, the paper cut off the sequences whose lengths are longer than 1022 codons. Could you please show the length distribution of the original collected dataset to prove that the training data covers the dominant sequence length range.
2. line 472 "Prot2RNA captures more nuanced synonymous codon preferences". What does "preferences" actually mean?
3. Would it be better to add mask indicative function into (1) and (2), like $1(x_i=M)$ in LlaDa paper?
4. In line 260, what does "less-to-more masking" strategy mean? does it mean less to more noise (masking)? It would be better to clearly state the direction of process, i.e., either forward diffusion process or backward diffusion process.

---

> ### Author Response · Authors · 2025-11-21
>
> We thank the reviewer for the careful reading of our manuscript and the extensive, constructive feedback. We appreciate the positive feedback regarding the dataset quality, modeling approach, and clarity of the manuscript. Below, we address each point in detail.
>
> **W1.** We appreciate the reviewer’s suggestion regarding autoregressive (AR) baselines. Our choice of a diffusion (masked-denoising) framework was motivated by the nature of the codon optimization task. Unlike natural language, mRNA sequences have no intrinsic left-to-right causality, and the optimal codon at each position depends on both upstream and downstream context (e.g., GC balance, folding constraints). The diffusion formulation allows parallel updates and bidirectional conditioning, which are well suited to these global dependencies. In contrast, AR models generate sequentially and tend to overfit local codon frequency biases. We have also revised Section 3.1 to make this rationale explicit in the main text.
> While models such as GenerRNA [1] or RNAGenesis [2] demonstrate AR generation for noncoding RNAs, adapting them to protein-conditioned mRNA generation would require substantial architectural and objective redesign (introducing codon-level tokenization and amino-acid conditioning). Such work is beyond the scope of this study but represents an interesting future direction.
>
> **W2.1** We agree that codon-level accuracy and similarity to highly expressed transcripts cannot be taken as evidence that Prot2RNA-generated sequences would themselves exhibit high expression and in the future revisions of the manuscript, we will clarify that Prot2RNA generates human-like coding sequences, and and try to evaluate expression level and other important biological properties with computational assessment methods and, ultimately, experimental validation.
>
> **W2.2** We appreciate the reviewer’s suggestion to include proxies such as Mean Ribosome Loading (MRL) and Translation Efficiency (TE). These are indeed valuable experimental measures of expression, but both depend primarily on the 5′ and 3′ untranslated regions (UTRs) that determine ribosome recruitment and initiation efficiency. Our work focuses exclusively on coding sequence (CDS) design, where synonymous codon choice affects elongation rate, mRNA stability, and folding, but not initiation. Because we do not generate or model UTRs, MRL and TE cannot be meaningfully computed for our generated sequences.
> Instead, we evaluate expression-related properties that are intrinsic to the CDS, including codon-level accuracy, Codon Adaptation Index (CAI), GC/GC3 composition, and MinMax codon-usage profiles, each known to correlate with translation efficiency and mRNA stability in human cells. These metrics provide indirect yet biologically interpretable proxies for expression potential within the coding region.
> Future extensions of Prot2RNA could jointly model UTRs or incorporate ribosome-profiling data, which would make direct evaluation with MRL or TE feasible.
>
> **W3.1** After re-evaluating the codon-level accuracy results, we agree that the small numerical difference does not constitute meaningful evidence of adaptation. In response, we have removed the sentence suggesting a benefit from fine-tuning and now report these values without interpretation.
>
> **W3.2** CAI is included as a descriptive metric rather than a target for optimization. As shown in prior biological literature, the most highly expressed human transcripts do not maximize CAI, and excessively high CAI can even negatively affect translation dynamics. We therefore evaluate whether Prot2RNA reproduces natural CAI distributions, not whether it increases CAI. We have added Appendix Section D (“Evaluation Metrics”) to the revised manuscript which explains all the used metrics in more detail and we hope this clarifies the answer.
>
> **W3.3** Our intention with the CAI and GC analyses was not to position them as functional metrics, but rather as sanity checks to verify that Prot2RNA generates sequences with natural human-like compositional properties. We agree that these metrics are limited and do not directly reflect functional optimization. In a future revision, we will streamline these comparisons and better emphasize the main strength of our approach: enabling protein-conditioned generation of human-like CDS sequences without handcrafted rules or curated codon tables. We appreciate the reviewer’s feedback and will use it to improve the clarity of the presentation.

---

> ### Author Response · Authors · 2025-11-21
>
> **W3.4** Sequence fidelity refers to how closely generated CDSs resemble the distributional properties of natural human coding sequences, including synonymous codon usage patterns, positional biases, GC and GC3 composition, MinMax profiles, and embedding-level similarity. Metrics such as codon accuracy, MinMax distance, GC/GC3, and FID-like embedding comparisons therefore fall under fidelity, and we do not want to interpret them as indicators of expression optimization.
> Relatedly, we agree that none of our reported metrics directly measure expression efficiency. During our work, we explored whether expression-related signals could be assessed computationally using several sequence-only predictors (fine-tuned CodonBERT [3,4], AIDO.RNA [5], and RiboNN [6]). These methods are trained to estimate distinct quantitative endpoints (transcript or protein abundance, translation efficiency) and their reported correlations with experimental measurements range 0.28–0.67, (depending on the task and dataset). When used on the ground-truth sequences in our dataset, the predicted values did not clearly separate the two groups (high and low expression), likely reflecting both the moderate predictive strength reported and the fact that these models quantify different biological properties (e.g., translation efficiency models depend strongly on features outside the CDS, such as the 5′UTR). For this reason, we refrained from interpreting such results and now emphasize that experimental validation is required to assess functional outcomes, which is explicitly stated as a limitation in the Conclusion.
> We would welcome any suggestions for computational evaluation strategies the reviewer considers biologically meaningful prior to eventual wet-lab validation.
>
> **Q1.** We thank the reviewer for the question. To address it, we now include in the Appendix Section A (Fig. 5) the CDS length distribution for the entire collected dataset, along with the distributions for (i) all sequences, (ii) all sequences that fall within the 1,022-token context window and (iii) the test set. These distributions show that the vast majority of human CDSs naturally fall within the 1,022-codon limit (>80%).
> During pretraining, random cropping is applied only to long single-modality sequences to ensure they fit within the context limit. During finetuning, however, no cropping is used: we simply select protein-CDS pairs whose full lengths fit the available window. Because the underlying CDS length distribution is broad and heavily weighted toward shorter transcripts, this filtering does not bias the training or test sets toward atypically short sequences.
> These analyses confirm that the model is trained and evaluated on CDS lengths that represent the typical and most common range found in the human transcriptome.
>
> **Q2.** We have clarified the wording to specify that “synonymous codon preferences” refer to context-dependent codon usage biases, as now stated in the revised text (Line 492).
>
> **Q3.** We thank the reviewer for carefully examining our equations. The formulation we use is equivalent in effect. While the LLaDA [7] paper includes an explicit mask indicator function and sums over all tokens (i=1…L), our formulation restricts the summation to masked positions only ($i \in \mathcal{M}$), where $\mathcal{M}$ is the index set of masked tokens as stated in the text. Thus, the masking indicator is implicitly represented by the summation domain.
>
> **Q4.** We thank the reviewer for pointing out that the description of the masking schedule could be clearer. We have revised the text in Section 3.4 (Sequence Generation) and Section 4.4 (Design Choice Analysis, under Decoding Strategies) to explicitly describe the direction of the process and the meaning of the less-to-more unmasking strategy. The updated text now clarifies that under the cosine schedule, an increasingly larger fraction of tokens becomes unmasked and refined at each iteration.
>
> ---
>
> [1] Zhao et al. (2024). GenerRNA: A generative pre-trained language model for de novo RNA design. PLoS One, 19(10), e0310814.
>
> [2] Zhang et al. (2024). RNAGenesis: A Generalist Foundation Model for Functional RNA Therapeutics. bioRxiv, 2024-12.
>
> [3] Li et al. (2024). CodonBERT large language model for mRNA vaccines. Genome research, 34(7), 1027-1035.
>
> [4] Li et al. (2025). mRNA-LM: full-length integrated SLM for mRNA analysis. Nucleic Acids Research, 53(3), gkaf044.
>
> [5] Zou et al. (2024). A large-scale foundation model for rna function and structure prediction. bioRxiv, 2024-11.
>
> [6] Zheng et al. (2025). Predicting the translation efficiency of messenger RNA in mammalian cells. Nature biotechnology, 1-14.
>
> [7] Nie et al. (2025). Large language diffusion models. arXiv preprint arXiv:2502.09992.

---

### Official Review · Reviewer_GGiz · 2025-11-05

**Soundness:** 2
**Presentation:** 2
**Contribution:** 2
**Rating:** 2
**Confidence:** 4

**Summary:**

The authors propose Prot2RNA, a diffusion language model for generating mRNA coding sequences conditioned on target proteins. The model is trained in two stages: masked diffusion pretraining on separate human protein and mRNA sequences, then finetuning on protein-mRNA pairs where the protein serves as an unmasked prompt. Sequences are generated through iterative denoising with learnable masking schedules. The authors evaluate on ~2,912 highly expressed test sequences using codon-level accuracy, CAI, GC content, and positional codon usage metrics, comparing against several baseline methods.

**Strengths:**

1. The authors curated and cleaned a large number of human protein-mRNA pairs from NCBI RefSeq and GENCODE with rigorous filtering criteria, and commit to making the dataset, code, and model weights publicly available, which will benefit the community.
2. Training exclusively on human coding sequences is well-motivated for the stated therapeutic mRNA applications.
3. The paper includes ablations examining model scale, positional encoding strategies, masking schedule functions, and number of denoising steps, providing insight into which architectural decisions matter.
4. Applying masked diffusion language models to protein-conditioned mRNA generation is a creative methodological contribution.

**Weaknesses:**

0. **No experimental validation of generated sequences.** The paper lacks any wet-lab experiments demonstrating that generated sequences actually improve protein expression, mRNA stability, or other functional properties. Wet-lab proof would be the ultimate evidence of the viability of the proposed approach. This is an expected limitation of the work that must be stated, however, I do not take it into account in my evaluation.


1. **Circular evaluation logic undermines validity.** The model is trained to mimic highly expressed sequences and evaluated by measuring similarity to highly expressed sequences, which tests memorization rather than whether the learned patterns genuinely improve expression. This circular reasoning provides no evidence that generated sequences would actually perform better than alternatives.


2. **Questionable choice of primary metric.** Codon-level accuracy (exact matching to wild-type) is treated as the key success metric, but there's no evidence that exactly replicating wild-type codons is optimal or that alternative synonymous codon choices couldn't achieve equal or better expression. The metric conflates similarity with quality.


3. **Inconsistent training paradigm lacks justification.** During pretraining the model receives either protein or RNA sequences separately, while during finetuning it processes concatenated protein-RNA pairs. This fundamental architectural inconsistency is not explained or justified, and no ablation studies compare against unified approaches (classic seq2seq models, concatenated pretraining throughout, ESM3-like joint pretraining, etc.).


4. **Insufficient explanation of evaluation metrics.** The paper uses multiple metrics (CAI, GC content, MinMax profiles, Frechet distance) with only brief name mentions and no detailed explanation of what they measure, how they are calculated, their biological significance, or why they are appropriate for evaluating codon optimization quality. The manuscript would greatly benefit of adding a section with detailed explanations and discussion on the metrics.


5. **Weak test set dissimilarity claims.** Clustering at 80% sequence identity still allows substantial similarity between training and test proteins, selecting "smallest clusters" may introduce systematic biases toward unusual or short sequences, and the characterization of test sequences as "sequentially dissimilar" is overstated.


6. **Misleading presentation of statistical rigor.** The paper reports results from a single run with fixed seed 42 (stated explicitly in Reproducibility Statement), yet presents box-and-whisker plots throughout that create an illusion of statistical robustness. These plots show inter-sequence variability across test examples from that single run, not inter-run variability across different random initializations, meaning there are no confidence intervals for model performance, no way to assess whether observed differences between methods are statistically significant versus artifacts of the particular random seed, and no evidence that results would replicate with different initializations.


7. **Inconsistent use of CAI.** The simple CAI-max baseline achieves codon accuracy (0.466) competitive with CodonBERT (0.477) and approaching CodonTransformer (0.502), yet this strong performance from a deterministic rule-based method raises unaddressed questions about whether the added model complexity is justified. At the same time, the authors cite evidence that CAI doesn't consistently correlate with expression (Vogel et al., 2010) yet still benchmark against CAI-maximization and uses CAI as an evaluation metric. This is an inadequate baseline comparison, or a fundamental tension about what actually matters for optimization.


8. **Absence of diversity and novelty analysis.** The model uses deterministic argmax decoding to generate exactly one sequence per protein, despite the introduction emphasizing the "vast design space". There is no exploration of multiple diverse high-quality candidates, no sampling strategies (temperature, top-k, nucleus), no measurement of inter-sequence diversity, and no assessment of whether generated sequences are novel or memorized from training data. This limits practical utility since therapeutic applications would benefit from multiple candidates for empirical testing, provides no way to validate whether the model learns underlying codon preference distributions versus memorizing single solutions, and offers no evidence about design space coverage or whether better alternatives exist nearby.


9. **Comparison against domain-expert-designed sequences or validated therapeutic constructs would strengthen the work.**


10. **Questionable Frechet distance metric design.** The proposed Frechet distance uses CodonBERT embeddings to measure semantic similarity, but if CodonBERT performs poorly at codon optimization (as the paper's own results suggest), why should its learned embeddings capture the relevant semantic properties for evaluation?


11. **Overfitting concerns from high accuracy.** Achieving high codon-level accuracy (>61%) with relatively similar train/test splits (90% vs 80% identity clustering) suggests the model may be memorizing training patterns rather than learning generalizable codon optimization principles.

**Questions:**

1. What sample sizes were used to calculate the Frechet distance, how was this justified as sufficient? Can the authors provide the reference FD value between the held-out test set and a random sample of the same size from the training data (preferably, computed multiple times for statistics) to establish a meaningful scale?

2. What justifies the median TPM > 5 cutoff for "highly expressed" sequences, how sensitive are results to this choice, and has this threshold been validated against expression in therapeutically relevant cell lines?

3. Can the authors provide CodonBERT results with its original post-processing enabled, since removing amino acid preservation safeguards may create an unfair disadvantage compared to other methods?

---

> ### Author Response · Authors · 2025-11-21
>
> We thank the reviewer for the careful reading of our manuscript and the extensive, constructive feedback. We appreciate the recognition of our dataset curation effort, the methodological contribution, and the usefulness of the ablation studies. Below, we address each point in detail.
>
> **W0.** We completely agree that wet-lab validation is the ultimate test of the functional impact of Prot2RNA-generated sequences. As noted, this is outside the scope of the current work but remains a core direction for future development. In the revised manuscript, we emphasize this point more explicitly by stating experimental validation as a limitation and the most important next step in the Conclusion section. We appreciate the reviewer’s understanding of this limitation and not taking it into account in their evaluation.
>
> **W1. & W2.** We agree that codon‐level accuracy and similarity to highly expressed transcripts cannot be interpreted as evidence that Prot2RNA-generated sequences themselves exhibit high expression. In future revisions of the manuscript, we will clarify that Prot2RNA generates human-like coding sequences, and that codon-level accuracy is intended as a naturalness/fidelity metric rather than an indicator of functional optimization.
> To explore whether expression-related signals could be assessed computationally, we evaluated several sequence-only predictors of expression and translation efficiency (including fine-tuned CodonBERT [1, 2], AIDO.RNA [3], and RiboNN [4]). These tools estimate different quantitative biological properties (e.g., transcript abundance, protein abundance, or even translation efficiency) and report moderate correlations with experimental measurements in their original publications (ranging between 0.28 and 0.671 for finetuned models, depending on the task and dataset). When we applied these models to the ground-truth sequences in our dataset, their predictions did not produce a clear separation between the two groups. This may reflect a combination of the moderate predictive correlations reported and the fact that these models target different biological properties (e.g., translation efficiency depends strongly on 5′UTR features, and RiboNN uses the whole sequence as input). Because these tools do not provide a consistent or unified evaluation signal in this setting, we avoid overinterpreting their outputs and instead emphasize experimental validation as the next essential step, which is now explicitly stated as a limitation in the Conclusion.
> We would welcome any suggestions for computational evaluation strategies the reviewer considers biologically meaningful prior to experimental validation.
>
> **W3.** We intentionally adopted a two-stage training paradigm for data-efficiency and representational reasons rather than architectural necessity. During pretraining, the model learns modality-agnostic sequence statistics from large, unpaired corpora of proteins and coding RNAs, which is infeasible with paired training only. Because the same transformer backbone and token vocabulary are used in both stages, the network weights remain consistent, only the masking strategy changes. Finetuning on paired protein-RNA data then aligns these pretrained representations for conditional generation. This structure mirrors the training strategy used in LLaDA [5], where the model is pretrained on unpaired text and only later finetuned on pairs for instruction-following (including translation from English to Chinese/German). We have revised Section 3.1 to make this rationale explicit in the main text.
> Seq2seq or encoder-decoder variants would constitute alternative architectures rather than ablations; our goal was to test whether masked diffusion alone suffices for protein-aware codon generation.
>
> **W4.** We have added two dedicated appendix sections:
> (1) Appendix Section D (“Evaluation Metrics”) that provides detailed descriptions of CAI, GC and GC3 content, and MinMax profiles, including how each metric is calculated; and
> (2) a separate section (Appendix Section G.1) explaining our Fréchet distance metric in greater depth with additional supporting experiments.

---

> ### Author Response · Authors · 2025-11-21
>
> **W5.** We thank the reviewer for this comment and appreciate the opportunity to clarify our intent. Protein LM benchmarks such as ESM-Cambrian [6], ESM [7], and AlphaFold 3 [8] typically cluster at 40–70% sequence identity because their tasks (structure prediction, homology modeling, and contact prediction) require distinguishing fine-grained evolutionary and structural signals. Our task is fundamentally different: we evaluate protein-conditioned codon generation, where the goal is to assess whether the model can produce plausible coding sequences for novel amino-acid sequences not seen during training.
> For this purpose, clustering at 80% identity is both appropriate and conservative: proteins with 80% identity typically differ at ~20% of residues, and because the mapping from protein → CDS is highly underdetermined, codon-level novelty is much higher than protein-level similarity (synonymous codons, context-dependent biases, and global composition constraints all vary independently of protein identity). Thus, even “similar” proteins at the amino-acid level correspond to very different CDS sequences.
> Regarding terminology, we have revised the manuscript to avoid the phrase “sequentially dissimilar,” which could imply high evolutionary divergence. Instead, we now describe the test set as “non-redundant at 80% protein sequence identity”, which more precisely reflects our clustering objective.
> Regarding the selection of the smallest clusters, this choice was made to preserve as much data as possible for training while ensuring that all test proteins are non-redundant with respect to the training set. Small MMseqs2 clusters do not correspond to “unusual” or “short” proteins; they simply contain proteins that are less represented in the human proteome and therefore have fewer close homologs. These proteins are still biologically typical (just less redundant) and represent exactly the kind of novel proteins for which we would like to assess generalization.
> To confirm that this selection does not skew the test set toward truncated sequences, we include the CDS length distribution of the test set and the full dataset in Figure 5 in Appendix Section A; the distributions cover the same biologically realistic range, with no enrichment for extremely short proteins.
>
> **W6.** Our model is deterministic, meaning given a target protein we will always get the same CDS sequence each time. If the comment was to train the model with different random initializations, we would have to pretrain it several times. Since pretraining the large language models is computationally expensive and timely (it took us two weeks to pretrain and finetune our model with the 2048 context length), we did not have enough time and resources to repeat the same procedure several times. We appreciate the comment and will try to repeat at least the finetuning part several times.
>
> **W7.** Our intention is not to present CAI as a direct predictor of expression. Instead, we use it as a compositional sanity check to ensure that generated sequences remain within biologically plausible ranges and do not over-optimize GC3% or collapse to extreme codon frequencies.
> The CAI-max baseline is included as a historical and methodological reference point, representing the extreme case of greedy codon selection. Prior codon optimization tools sometimes report CAI-max versions as baseline sequences (e.g., Trias), which allows contextualizing our method against legacy heuristics.
> Importantly, Prot2RNA does not maximize CAI. Our generated sequences achieve moderate CAI values closer to natural high-expression transcripts, aligning with the well-established observation that excessive CAI can impair mRNA stability or folding. We have added Appendix Section D (“Evaluation Metrics”) to the revised manuscript which explains all the used metrics in more detail.
>
> **W8.** While we did not apply temperature, top-k, or nucleus sampling, we did generate multiple sequences per protein using different masking schedules (uniform, cosine, square-root, cubic) and different numbers of denoising iterations, which already produce distinct variants under controlled decoding settings. These variants remain within biologically plausible boundaries (GC/GC3/CAI ranges; codon-level fidelity; MinMax patterns), which is consistent with the goal of generating human-like CDSs.
> A dedicated analysis of inter-sequence variation, and novelty relative to the training distribution is an interesting extension and a natural direction for future work. We would welcome any suggestions for specific diversity metrics or protocols that the reviewer finds most informative for this task.

---

> ### Author Response · Authors · 2025-11-21
>
> **W9.** We thank the reviewer for this suggestion. Following prior work, we compared Prot2RNA against sequences generated by two widely used commercial codon-optimization services (Genewiz and IDT). Because these tools require per-sequence online submissions and have long processing times, we evaluated a representative subset of 200 proteins from our test set selected to match the distribution of protein lengths.
> The results (Appendix Section G.2) show that both commercial tools generate sequences that diverge more strongly from human wild-type CDS patterns, particularly in codon-level accuracy, where Prot2RNA achieves substantially higher agreement. In terms of composition, Genewiz tends toward higher CAI and elevated GC/GC3 values, while IDT produces more moderate compositions. Prot2RNA remains closest to wild-type distributions across these metrics, indicating that it captures human-like codon-usage characteristics rather than optimizing primarily for frequency-based heuristics.
> We emphasize that this comparison is not intended to replace biological validation. As also added in our conclusion, wet-lab evaluation remains an important next step, and these results should be viewed as complementary evidence alongside our main in-silico evaluation. At present, comparing to commercial codon-optimization tools provides the most practical external reference point for assessing the properties of Prot2RNA-generated sequences.
>
> **W10.** The “CodonBERT” [1] used to compute the Fréchet distance is not the same model as the CodonBERT [9] baseline evaluated for codon optimization. We use the encoder-only CodonBERT pretrained by Sanofi solely as a frozen feature extractor to obtain biologically meaningful sequence embeddings, analogous to how Inception features are used in FID for images. This encoder was trained on millions of natural coding sequences for representation learning, not on optimization tasks, and thus provides a stable semantic embedding space for comparing distributions of coding sequences.
> We have revised the text to make this distinction explicit and to clarify that this evaluation uses a fixed pretrained encoder independent of any generative baseline.
>
> **W11.** As discussed in our response to Comment W5, protein-level similarity does not translate into CDS-level redundancy: even proteins sharing 80% identity typically diverge strongly at the codon level due to synonymous variability and context-dependent biases. As a result, high protein identity does not permit trivial memorization of CDS sequences.
> The observed codon-level accuracy therefore reflects the model’s ability to learn stable, global codon-usage patterns rather than recall specific CDSs from the training set. This is further supported by the fact that codon choices for identical amino acids vary substantially across the human transcriptome, making memorization an ineffective strategy for achieving high accuracy on unseen protein sequences.
> We agree that deeper analyses of codon-level novelty (e.g., n-gram or motif-level divergence relative to the training set) would strengthen future work, and we view this as a natural extension of our evaluation framework.
>
> **Q1.** We have added a dedicated appendix subsection (Appendix G.1: Fréchet Distance Evaluation) that analyzes the sample-size dependence of the Fréchet distance, reports the corresponding noise floor, and includes large-sample (10k) evaluations for our models. We believe this addition fully addresses the question of how sample size was chosen and how to interpret the reported values.
>
> **Q2.** We adopted the TPM>5 criterion directly from CodonBERT [9], where this threshold was chosen empirically because it corresponds to the upper tail of the TPM distribution in their HPA-derived dataset (top 5%). We did not explore alternative TPM thresholds, as this would require recomputing the cluster-based train/validation/test splits and retraining the entire model, which is outside the scope of the current study.
> The TPM values we use come from the Human Protein Atlas [10] and describe transcript abundance across healthy human tissues, which matches our goal of learning protein-mRNA relationships under normal physiological conditions. These data types are also consistent with common mRNA-therapeutic contexts, where model design typically assumes healthy baseline cellular states.
> Future work could focus on specific diseases or target tissues and adapt expression-based filtering to the particular cellular environment relevant for those applications.

---

> ### Author Response · Authors · 2025-11-21
>
> **Q3.** We thank the reviewer for this suggestion. We evaluated CodonBERT [9] with its original post-processing pipeline enabled (“CodonBERT_OG”), which restores amino-acid preservation safeguards. The results differ only marginally from the version we reported in the main text:
> Codon-level accuracy: 0.477 (CodonBERT) → 0.479 (CodonBERT_OG)
> GC content: 0.6631 → 0.6635
> CAI: unchanged at 0.974
> CodonBERT_OG performs nearly the same as the non-postprocessed version, and both remain well below CodonTransformer, Trias, and Prot2RNA variants.
> Because the effect size is extremely small and does not alter any conclusions or model ranking, we do not believe an additional figure or expanded table would add clarity to the manuscript.
>
> ---
>
> [1] Li, S., Moayedpour, S., Li, R., Bailey, M., Riahi, S., Kogler-Anele, L., ... & Jager, S. (2024). CodonBERT large language model for mRNA vaccines. Genome research, 34(7), 1027-1035.
>
> [2] Li, S., Noroozizadeh, S., Moayedpour, S., Kogler-Anele, L., Xue, Z., Zheng, D., ... & Jager, S. (2025). mRNA-LM: full-length integrated SLM for mRNA analysis. Nucleic Acids Research, 53(3), gkaf044.
>
> [3] Zou, S., Tao, T., Mahbub, S., Ellington, C. N., Algayres, R., Li, D., ... & Xing, E. P. (2024). A large-scale foundation model for rna function and structure prediction. bioRxiv, 2024-11.
>
> [4] Zheng, D., Persyn, L., Wang, J., Liu, Y., Ulloa-Montoya, F., Cenik, C., & Agarwal, V. (2025). Predicting the translation efficiency of messenger RNA in mammalian cells. Nature biotechnology, 1-14.
>
> [5] Nie, S., Zhu, F., You, Z., Zhang, X., Ou, J., Hu, J., ... & Li, C. (2025). Large language diffusion models. arXiv preprint arXiv:2502.09992.
>
> [6] ESM-Team. ESM Cambrian: Revealing the mysteries of proteins with unsupervised learning. EvolutionaryScale Website, https://evolutionaryscale.ai/blog/esm-cambrian (2024).
>
> [7] Lin, Z., Akin, H., Rao, R., Hie, B., Zhu, Z., Lu, W., ... & Rives, A. (2023). Evolutionary-scale prediction of atomic-level protein structure with a language model. Science, 379(6637), 1123-1130.
>
> [8] Abramson, J., Adler, J., Dunger, J., Evans, R., Green, T., Pritzel, A., ... & Jumper, J. M. (2024). Accurate structure prediction of biomolecular interactions with AlphaFold 3. Nature, 630(8016), 493-500.
>
> [9] Ren, Z., Jiang, L., Di, Y., Zhang, D., Gong, J., Gong, J., ... & Ni, M. (2024). CodonBERT: a BERT-based architecture tailored for codon optimization using the cross-attention mechanism. Bioinformatics, 40(7), btae330.

---

### Meta-Review · Area_Chair_yZww · 2025-12-30

**Summary:**

While the high-quality, well-curated human protein–mRNA dataset is valuable and the adaptation of diffusion language models to protein-conditioned CDS generation is technically sound, the core empirical concerns remain unresolved. The evaluation relies almost entirely on similarity-based metrics that cannot verify true expression or functional improvement. As a result, the effectiveness of the proposed model cannot be convincingly validated. Therefore, I lean toward rejecting this paper.

**Reviewer Concerns:**

This paper raises many concerns across. Most reviewers listed more than 10 issues, many of which affect the core validity of the claims.

**Concerns Largely or Partially Addressed**

- Metric definitions and clarity: The authors added detailed appendices explaining CAI, GC/GC3, MinMax profiles, and the Fréchet distance, and clarified terminology. This addresses clarity concerns raised by Reviewers GGiz and jC16, but does not resolve the fundamental issue that these metrics are weak surrogates for biological function.

- In response to Reviewer jc16, the authors justified the choice of diffusion model over AR model, arguing that mRNA folding and global constraints such as GC balance benefit from bidirectional context, which AR models struggle to preserve. They also explained the advantages of diffusion models compared with non-diffusion alternatives.

- The authors reasonably justified pretraining on unpaired corpora followed by finetuning on paired data as a data-efficiency strategy, drawing parallels to LLaDA-style training.

- New comparisons against commercial codon-optimization tools (Genewiz and IDT) were added. These show that Prot2RNA generates sequences closer to human wild-type patterns than heuristic services.

**Outstanding Concerns**

- The central concern from most reviewers is the lack of evidence for true expression improvement. It remains unresolved. Metrics such as codon accuracy and CAI do not imply actual high expression. There is no evidence that sequences resembling highly expressed natural mRNAs will themselves be novel, high-expressing mRNAs when synthesized. The rebuttal explicitly acknowledges this limitation.

- The authors acknowledge that results are obtained from a single fixed seed and that the model is deterministic. They cannot provide statistical significance for the results.

- The authors did not provide a rigorous analysis of whether the model generates novel codon solutions or just recalls motifs from the training data.

**Reviewer Scores:**

Reviewer GGiz (Rating: 2)
> Unlikely to change. The rebuttal confirms the reviewer’s core concerns regarding circular evaluation, weak biological claims, and lack of statistical rigor.

Reviewer jC16 (Rating: 4)
> Likely remain at 4 or slightly more positive. Several clarity issues were addressed, including removal of the weak finetuning adaptation claim and clearer diffusion vs. AR justification. However, ambiguity about the paper’s goal (sequence fidelity vs. expression optimization) and the lack of stronger validation remain.

Reviewer SWB8 (Rating: 2)
> Unlikely to change. This reviewer emphasized the absence of biologically grounded evidence. The authors’ reframing of the work as “fidelity-focused” does not address the reviewer’s demand for functional validation.

Reviewer aGVJ (Rating: 2)
> Likely unchanged. The reviewer explicitly stated that their main hesitation remains, as all reported metrics are still surrogates for biological relevance.

---

### Decision · Program_Chairs · 2026-01-26

Reject